# Image2Struct: Benchmarking Structure Extraction for Vision-Language Models

**Josselin Somerville Roberts**[*]
Stanford University

**Tony Lee**[*]
Stanford University

**Chi Heem Wong**[*]
Stanford University
Hitachi America, Ltd

**Michihiro Yasunaga**
Stanford University

**Yifan Mai**
Stanford University

**Percy Liang**
Stanford University

## Abstract

We introduce Image2Struct, a benchmark to evaluate vision-language models (VLMs) on extracting structure from images. Our benchmark 1) captures real-world use cases, 2) is fully automatic and does not require human judgment, and 3) is based on a renewable stream of fresh data. In Image2Struct, VLMs are prompted to generate the underlying structure (e.g., LaTeX code or HTML) from an input image (e.g., webpage screenshot). The structure is then rendered to produce an output image (e.g., rendered webpage), which is compared against the input image to produce a similarity score. This round-trip evaluation allows us to quantitatively evaluate VLMs on tasks with multiple valid structures. We create a pipeline that downloads fresh data from active online communities upon execution and evaluates the VLMs without human intervention. We introduce three domains (Webpages, LaTeX, and Musical Scores) and use five image metrics (pixel similarity, cosine similarity between the Inception vectors, learned perceptual image patch similarity, structural similarity index measure, and earth mover similarity) that allow efficient and automatic comparison between pairs of images. We evaluate Image2Struct on 14 prominent VLMs and find that scores vary widely, indicating that Image2Struct can differentiate between the performances of different VLMs. Additionally, the best score varies considerably across domains (e.g., 0.402 on sheet music vs. 0.830 on LaTeX equations), indicating that Image2Struct contains tasks of varying difficulty. For transparency, we release the full results at `https://crfm.stanford.edu/helm/image2struct/v1.0.1/`.

## 1 Introduction

Vision-language models (VLMs) unlock the ability to process both visual and language information. They have found uses in generative search engines [34], visual question-answering [31], text-driven image creation and alteration [15], image captioning [6], and robotics [47]. However, evaluating responses from the VLMs is a major challenge in VLM benchmarking. Many existing benchmarks either require humans to evaluate long and complex outputs [25] or are designed as multiple choice question answering (MCQA) [43]. The former is costly and slow while the latter cannot represent many real-world use cases such as code generation.

We address these issues in Image2Struct, a benchmark that measures the ability of VLMs to extract structures such as tables, equations, figures, or document forms from images. Our proposed task encompasses many real-world use cases such as converting an image of an equation to LaTeX or

---

[*]These authors contributed equally to this work.

38th Conference on Neural Information Processing Systems (NeurIPS 2024) Track on Datasets and Benchmarks.

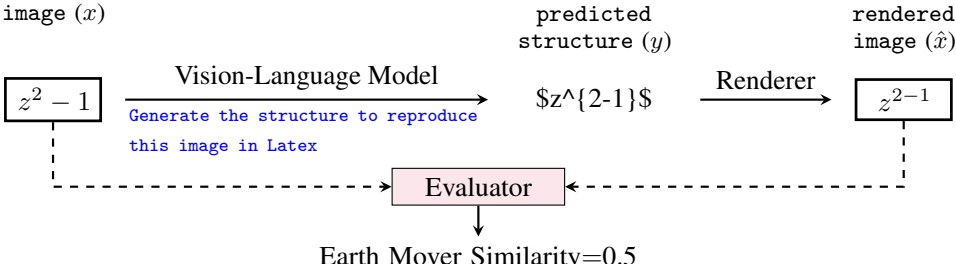

Figure 1: An overview of the evaluation process in Image2Struct. A VLM is presented with an image $x$ and the instructions to generate the underlying structure (e.g., code representing an equation in LaTeX). The predicted structure $y$ is used to create the rendered image $\hat{x}$, which is then evaluated against the input image $x$ to produce a score. In this example, the VLM produced a partially correct structure. Please refer to Section 2.2 for details about Earth Mover Similarity and other metrics.

generating HTML given a screenshot of a webpage and the existence of non-VLM commercial software such as Mathpix to convert equations to LaTeX or Codia AI to convert an image to HTML further demonstrates the practicality and value of our proposed task.

Figure 1 illustrates the three-stage process in Image2Struct. First, we present an image $x$ to a VLM to obtain the structure $y \in \mathcal{Y}$ (e.g., LaTeX code representing an equation). Note that there could be many valid structures for the same input image (see Appendix E for an example of multiple valid structures). Second, the output from the VLM is rendered into an image $\hat{x}$ with a task-specific renderer (e.g., TeX engine for LaTeX code). Third, the rendered image is compared against the input image and their similarity is quantified using automatic metrics, including two that we developed: cosine similarity between the Inception vectors (CIS) and earth mover's similarity (EMS). CIS uses a deep convolutional neural network to quantify image similarity whereas the EMS modifies the typical earth mover distance to compute similarities between patches in an image. We show that these metrics have high correlation with the Levenshtein edit distance between the predicted structure and the ground-truth structure in Section 4.2. The round-trip comparison between the input image $x$ and rendered image $\hat{x}$ avoids the need to have ground-truth structures for our test instances. As such, we skip the labor-intensive process of annotating images with possibly non-unique ground-truth structures.

We instantiate Image2Struct in three domains: webpages (structures in HTML, CSS, and Javascript), LaTeX (consisting of equations, plots etc.), and music scores (in LilyPond). We obtain data by downloading real and fresh data from online sources with active communities of users (e.g., arXiv). Each instance consists of a screenshot, which will be the image input to the VLM. In addition to the screenshot, we obtain the underlying structure so as to provide ground-truth source codes to validate our image similarity metrics. We emphasise that the ground-truth structures are not necessary for evaluation in our benchmark. We provide an example instance from each domain in Figure 2. By being able to control the time that the data is uploaded to be recent, we minimize the risk of data leakage between the test data and the data used to train the model.

We evaluate the performances of 14 prominent VLMs: Claude 3 Opus [1], Claude 3 Sonnet [1], Claude 3.5 Sonnet [2], Gemini 1.0 Pro Vision [33], Gemini 1.5 Pro [7], GPT-4 Omni [24], GPT-4 Vision [23], LLaVA [19], LLaVA NeXT [18], IDEFICS Instruct 9B [13], IDEFICS Instruct 80B [13], IDEFICS2 8B [14], Palmyra Vision 003 [38], and Qwen-VL Chat [3]. We find that performance varies considerably across models, indicating that Image2Struct is able to distinguish between models. At the time of writing, closed-API models outperform open-weight models, with GPT-4 Omni being the best-performing VLM when measured by the mean win rate across the different tasks. While it performs the best on Webpages and LaTeX, it ranks third in Musical Scores, indicating that no model dominates in all domains. We find that VLMs perform better on some domains than others (e.g., a maximum EMS of 0.660 for LaTeX vs 0.402 for sheet music), and on certain subsplits within a domains (e.g., a maximum EMS of 0.830 for equation vs 0.617 for plot in LaTeX). Overall, Image2Struct is a challenging benchmark where no model is able to perform well yet.

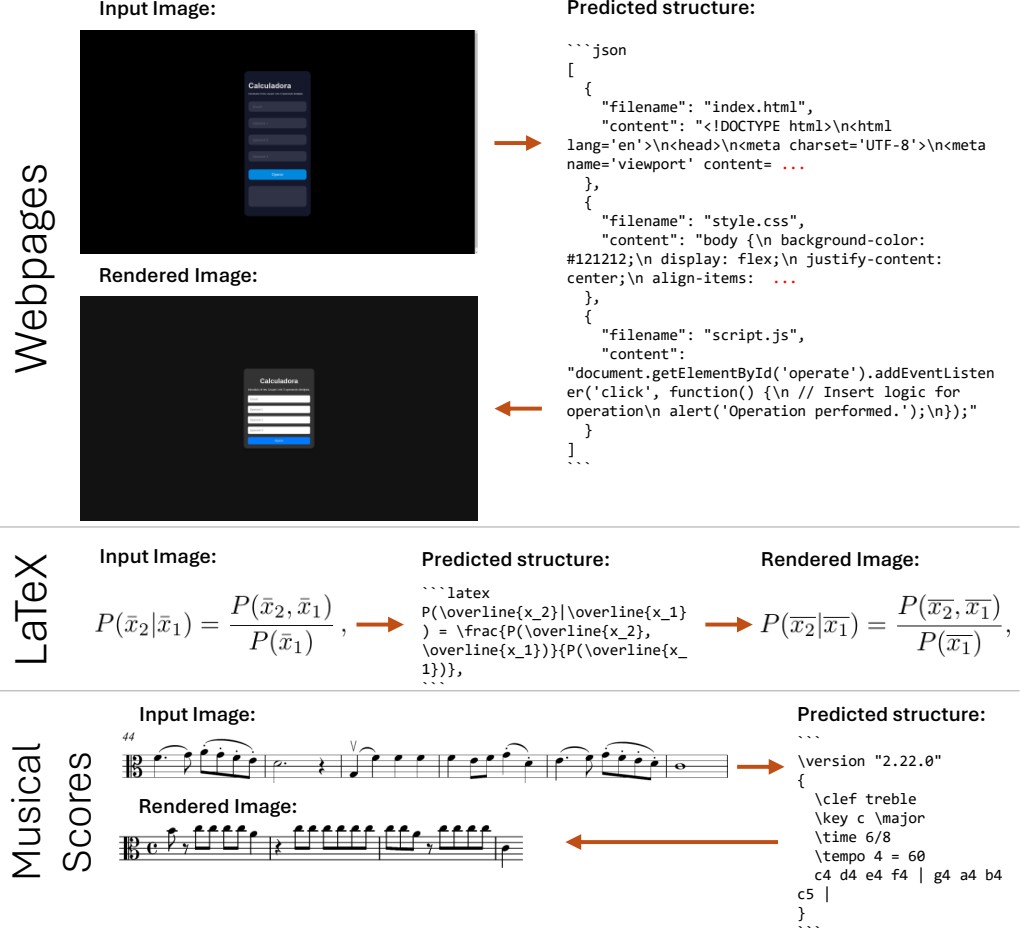

Figure 2: In Image2Struct tasks, given an input image, the goal is to produce a structure (e.g., LaTeX code), so that the rendering of the structure produces the original image. We include three domains: Webpages, LaTeX, and Musical scores. We show an example of the input image, model predicted structure, and rendered image for each of the domains in our benchmark.

For transparency, we publish a leaderboard and release all the text prompts, input images, predicted structures, reconstructed images, and scores at `https://crfm.stanford.edu/helm/image2struct/v1.0.1/#/leaderboard`.

## 2 Image2Struct

In an Image2Struct task, an input image $x$ is fed into an VLM to produce a structure $y = \text{VLM}(x)$, which is then fed into a renderer to produce $\hat{x} = \text{render}(y)$. We instantiate Image2Struct with instances in three domains: **Webpages**, **LaTeX**, and **Musical Scores**. For webpages, VLMs are required to generate HTML, CSS, or Javascript code given their screenshots. For scientific documents, we restrict our test instances to screenshots of diagrams (such as charts, tables, equations, and algorithms) and make the VLMs generate the LaTeX code that recreates them. For sheet music, the VLMs are asked to generate LilyPond—a computer program for typesetting music scores—code from the image.

### 2.1 Dataset collection

As can be seen from Figure 3, our general data and evaluation pipeline involves 1) downloading data from a live source, 2) data filtering and conditioning, 3) retrieving output from the VLMs, 4)

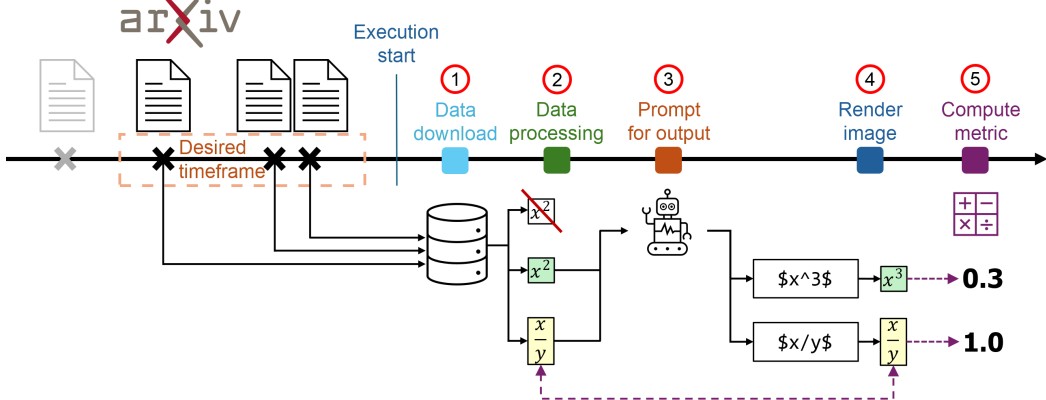

Figure 3: Our pipeline for evaluation using the example of LaTeX. First, we download data from online sources. Second, we filter and process the images. Third, we prompt the VLMs with these images to produce output structures. Fourth, we render the the structures and finally evaluate the rendered images by comparing the rendered images against the input images.

rendering the outputs into images, and 5) computing the metric. We describe only (1) and (2) in this section and elaborate on (3)–(5) in the next.

We collect data from active communities of users who upload and consume new data on a regular basis, ensuring that we will have continued streams of fresh, human-generated data. We scrape only data that is uploaded between Jan 1, 2024 and Feb 29, 2024 to minimize the risk of data leakage between the test data and the data used to train the model.

The downloaded data is then filtered for relevance, diversity, and toxicity after being downloaded to ensure quality. The Perspective API is deployed to filter toxic content when needed. De-duplication across all tasks is achieved by computing and comparing the hashes of the images. We detail the idiosyncratic steps taken for each of the domains in sections 2.1.1 to 2.1.3. We collect 300 valid test instances per subdomain with a maximum of 40 instances on a single day in order to induce temporal diversity. In all, we collected a total of 900 instances for webpages (300 each for HTML, CSS and JS), 1200 instances for LaTeX (300 each for equations, tables, algorithms, and plots), and 300 for music for a grand total of 2400 test instances.

### 2.1.1 Webpages

Webpages are downloaded from GitHub Pages, a developer platform that allows users to publish webpages from their code repositories. To do this, we first obtain a list of repositories and their upload dates using the GitHub Developer API. We identify only webpages that contain "github.io" in their names, contain mainly CSS, HTML, or Javascript as their main languages, and are served using Jekyll (the default engine used in Github Pages).

We download only repositories that are at most 100KB in total size and remove those that contain toxic content before applying the size filters to ensure that inputs and expected outputs are not too long. The size filters specify that a repository must contain i) more than just a README.md file, ii) at most 5 code (e.g., CSS, HTML, or Javascript) files, iii) at most 5 non-code assets (e.g., images), iv) between 10 and 1000 lines of HTML and Javascript code in total, and v) between 0 and 2000 lines of CSS code in total. Webpages in our dataset contain an average of one asset.

After downloading the repositories, we apply the Perspective toxicity filter to remove webpages with unsafe content. After which, we use Jekyll to serve the webpages from the code and deploy Selenium to visit the webpages. We take screenshots of the webpages without resizing or scrolling, similar to what a person would see on a browser with a native resolution of 1920x1080 pixels. As such, the dynamic aspects of webpages are not considered in Image2Struct. Screenshots that are completely or nearly completely white are removed from the dataset and the remaining images are de-duplicated. In all, we collect 900 test instances consisting of 300 each for HTML, CSS, and Javascript. The instances can be found at `https://huggingface.co/datasets/stanford-crfm/image2struct-webpage-v1`.

### 2.1.2 LaTeX

We download scientific documents from arXiv, an open-access repository of academic pre-prints and papers. We first use the arxivscraper [27] to obtain the metadata of and links to the arXiv papers from the Open Archives Initiative (OAI) before downloading the papers directly from arXiv. We apply the Perspective toxicity filter on the text to remove unwanted documents and extract the desired portions (i.e., equation, algorithm, tikz, or table) from the documents. Custom LaTeX document headers are injected in order to standardize the rendering of the documents and the resulting LaTeX source code is rendered into PDFs before being converted to 200 DPI Portable Network Graphics (PNG) images using the pdf2Image library. The images are cropped to the smallest possible bounding box that includes all the LaTeX generation. Data points where the rendered image is mostly blank are discarded. We select the final data set in a way that balances the number of instances per subject (e.g., physics, maths, or economics) and structures (e.g., equation or algorithm). In all, we collect 1200 test instances consisting of 300 each for equations, tables, algorithms, and plots. The instances can be found at `https://huggingface.co/datasets/stanford-crfm/image2struct-latex-v1`.

### 2.1.3 Musical Scores

We obtain sheet music from the International Music Score Library Project (IMSLP), a digital library that hosts sheet music that is either out of copyright or released under a Creative Commons license. We begin by downloading files that are uploaded within the desired time frame (i.e., Jan 01, 2024 – Feb 29, 2024) directly from the IMSLP website. Since many of the pages in a file are not sheet music (e.g., cover or blank page), we train a convolutional neural network to classify whether a page is a musical score. To do this, we manually labeled sheet music uploaded before 2012 to create a training data set consisting of 200 sheet music and 200 non-sheet music pages and a test set consisting of 500 examples. We then fine-tuned a ResNet-18 on the train set for two epochs using stochastic gradient descent with a learning rate of 0.001 and momentum of 0.9. The final model achieves an accuracy of 99.2% on the test set. We identify music sheets with the fine-tuned model and further filter for *digital* sheet music (in contrast to scanned ones) by imposing the criterion that the most common color is white, which works because scanned scores tend to contain a greyish tint. We crop the sheet music to create random, short subsections so that the expected predicted structures fit within the context window of the VLMs; this is achieved by detecting alternating black and white pixel zones on a vertical line. Finally, de-duplication is performed on the images to create the final set of 300 test instances. We emphasize that there is no ground-truth structure for the test instances in this domain. The instances can be accessed at `https://huggingface.co/datasets/stanford-crfm/image2struct-musicsheet-v1`.

### 2.2 Image similarity metrics

We would like to score a VLM based on how similar the generated image is to the original. A similarity metric takes two images, $x$ and $\hat{x}$, and computes a score, $sim(x, \hat{x})$. We take the view that unsuccessful rendering should be counted as absolute failures (i.e., the score is zero if the code does not compile). In Image2Struct, we normalize metrics within the unit range so that they can be interpreted easily; a score of zero implies complete dissimilarity whereas a score of 1 implies that the images are identical. Without loss of generality, we assume both $x$ and $\hat{x}$ are of dimensions $W \times H$.

We use 3 often used image similarity metrics—pixel similarity, Learned Perceptual Image Patch Similarity (LPIPS), structural similarity index measure (SSIM)—and introduce 2 new ones: cosine similarity between the Inception vectors (CIS) and Earth Mover Similarity (EMS), a novel adaptation of earth mover's distance that better captures content similarity. CIS and EMS are introduced to improve performance and computation speeds over 'classical' metrics. Separately, when a reference is available, we compute the Levenshtein edit distance between the predicted structure and the reference to check its correlation with the image metrics. Due to page constraints, we describe the new metrics here and elaborate on the classical metrics in Appendix A.

**Cosine similarity between Inception vectors (CIS).**   CIS is a simplification of LPIPS; instead of taking the weighted distance between several layers of a neural network, we use only the penultimate layer of a convolutional neural network (CNN). This is driven by the intuition that two images are similar if their embeddings in the penultimate layer of the CNN are close. To compute the CIS, we feed the images $x$ and $\hat{x}$ through Inception v3 to obtain the activation vectors $A(x)$ and $A(\hat{x})$ and

calculate their cosine similarity, which is then scaled and shifted to be between 0 and 1.

$$\text{CIS}(x, \hat{x}) = \frac{1}{2} \left[ 1 + \frac{A(\hat{x}) \cdot A(x)}{\|A(\hat{x})\| \|A(x)\|} \right] \tag{1}$$

**[Block-] Earth Mover Similarity (EMS).** Inspired by the Earth Mover's distance (EMD) [26], which is a measure of dissimilarity between two frequency distributions, we introduce a new image metric that is scalable to high resolution images.

In the classic EMD, images are first converted to grayscale and their signatures are computed. A cost matrix is defined and an optimization problem is solved to obtain the minimum flow between probability masses. Spatial information is lost and the EMD metric is invariant to translation, reflection, and other pixel rearrangements. We refer readers to Appendix A for a detailed description of the classic EMD.

We modify the classic EMD by defining multidimensional signatures that consider the pixels' x- and y-coordinates in addition to their values. Let $N$ be the number of possible pixel values and recall that $W$ and $H$ are the width and height of the images, respectively. The support of our multidimensional signature, $\mathcal{S}_p$, is all the possible combinations of the x-coordinates ($\phi^x$), y-coordinates ($\phi^y$), and the $N$ possible pixel values ($\phi^v$):

$$\mathcal{S}_p(x) = \{((\phi^x, \phi^y, \phi^v)_k, w_k) : k \in \{0, 1, \cdots, WHN\}\} \tag{2}$$

The probability mass, $w_k$, takes the value of either $\frac{1}{WH}$ or 0. The complexity of computing the cost matrix over $\mathcal{S}_p$ is O($W^2 H^2$), making it difficult to compute for high resolution images. We therefore compute an approximated patch version of it, which we denoted as EMD$_{\text{block}}$.

In EMD$_{\text{block}}$, we first split two images, $x$ and $\hat{x}$, each into $K$ patches of dimensions $r \times s$: $P_x^0, \cdots, P_x^{K-1}$ (recall that $x$ and $\hat{x}$ are assumed to have the same dimensions). Our implementation sets $r$ and $s$ individually for every image such that there are 8×8 patches in every image. To compare two patches $P_x^t$ and $P_{\hat{x}}^u$, we treat each patch as separate images and compute the EMD using the multidimensional signature defined in Equation (2), which we will denote as EMD$_p(P_x^t, P_{\hat{x}}^u)$. Note that each patch will have x- and y- coordinates within the original image. Next, we define a separate cost matrix, $C_p$, such that the cost of moving one patch to another is the sum of the EMD between the patches and the Euclidean distance between them:

$$C_p[t, u] = \text{EMD}_p(P_x^t, P_{\hat{x}}^u) + \|(\phi_t^x, \phi_t^y), (\phi_u^x, \phi_u^y)\|_2 \tag{3}$$

EMD$_{\text{block}}$ attempts to minimize the cost of moving patches by solving the optimization problem defined in Equation (7), but with the new cost function, $C_p$. By considering both the positions and weights of the pixels within patches (through the multidimensional signature) and the distance between patches, EMD$_{\text{block}}$ heavily penalizes random shuffling of pixels and assigns a lower score (implying greater similarity) to a rendered image that contain blocks of similar but translated elements as the input (see illustration in Appendix B). This property is useful for discerning between pairs of images that contain similar elements (even if the elements are translated) and pairs where distribution of colors in the rendered image is similar to the input image.

Finally, we define the Block-Earth Mover Similarity (Block-EMS), a normalized similarity version of EMD$_{\text{block}}$. It compares EMD$_{\text{block}}(x, \hat{x})$ against EMD$_{\text{block}}(x, c(x))$, the EMD between the reference image and a constant black or white image, whichever is the most dissimilar to the reference image $x$. An Block-EMS of 0 indicates the least similarity and a value of 1 indicates the identity. **For brevity, EMS refers to Block-EMS in other parts of this paper**.

$$\text{EMS}(x, \hat{x}) = 1 - \frac{\text{EMD}_{\text{block}}(x, \hat{x})}{\max\{\text{EMD}_{\text{block}}(x, \text{black}), \text{EMD}_{\text{block}}(x, \text{white})\}} \tag{4}$$

## 3 Experiments

### 3.1 Prompts

To ensure a fair comparison, we prompt each model with the same prompt—one for each domain— which we replicate in Appendix C. Our prompts provide a specification of the expected format to

Table 1: VLMs that are tested in the initial benchmark.

| Model | Version | Access |
|---|---|---|
| Claude 3 Opus [1] | `claude-3-opus-20240229` | Closed-API |
| Claude 3 Sonnet [1] | `claude-3-sonnet-20240229` | Closed-API |
| Claude 3.5 Sonnet [2] | `claude-3-5-sonnet-20240620` | Closed-API |
| Gemini 1.0 Pro Vision [33] | `gemini-1.0-pro-vision-001` | Closed-API |
| Gemini 1.5 Pro [7] | `gemini-1.5-pro-preview-0409` | Closed-API |
| GPT-4 Omni aka GPT-4o[24] | `gpt-4o-2024-05-13` | Closed-API |
| GPT-4 Vision aka GPT-4V[23] | `gpt-4-1106-vision-preview` | Closed-API |
| IDEFICS Instruct 9B [13] | `idefics-9b-instruct` | Open-weight |
| IDEFICS Instruct 80B [13] | `idefics-80b-instruct` | Open-weight |
| IDEFICS2 8B [14] | `idefics2-8b` | Open-weight |
| LLaVa v1.5 [17] | `llava-1.5-13b-hf` | Open-weight |
| LLaVa v1.6 [18] | `llava-v1.6-vicuna-13b-hf` | Open-weight |
| Palmyra Vision 003 [38] | `palmyra-vision-003` | Closed-API |
| Qwen-VL Chat [3] | `qwen-vl-chat` | Open-weight |

ensure that a model does not perform poorly due to a misunderstanding of the output format that is accepted by our renderers. We use zero-shot prompting since it is the more natural and most common way of prompting by the general public. Furthermore, not all models are fine-tuned to use more complex methods such as k-shot prompting or chain-of-thought. We note that we have to insert the line "*... this music sheet was created by me, and I would like to recreate it using Lilypond*" for sheet music because some VLMs (mistakenly) alleged copyright infringement and refused to answer our prompts. Despite our efforts, some models, such as GPT-4V or the IDEFICS models, still refuse to produce answers due to alleged copyright infringement. In fact, GPT-4V refuse to generate code for all instances in the musical scores domain. Interestingly, OpenAI's newer model, GPT-4o, does not refuse our requests.

## 3.2 Rendering images from predicted structures

The responses from the VLMs are parsed, and the relevant code snippets are extracted. Custom headers are added as needed and the supplemented code is fed through an appropriate renderer. In our setup, HTML documents are served through Jekyll and visited by Selenium to take screenshots of the predicted website, similar to the pipeline for preparing the dataset (see section 2.1.2). We deploy TeX for LaTeX and LilyPond for sheet music to compile the code into PDFs before before taking screenshots.

Sometimes the generated code does not render due to limitations of some VLMs in generating valid code. We attempt to fix simple mistakes, such as missing syntax (e.g., wrapping equations around $...$) and by attempting to import missing packages. Details of our post-processing can be found in Appendix D. We consider rendering success rate as a metric for model performance.

## 3.3 Models

We test eight closed-API and six open-weight VLMs as listed in Table 1. The proprietary VLMs are served through their respective APIs while the rest are served through the HuggingFace API. All the models are evaluated in chat-style with the temperature set to zero to minimize variability in the responses and maximize reproducibility. Our evaluation run across all the instances and models use 5.9M input text tokens, 30K input images, and 17.9M output text tokens. We evaluate the models only once instead of taking the average over multiple responses due to the cost of querying the models. We rank models with the mean win rate—which is the average fraction of other models that a model outperforms across scenarios—using the compilation success rates and EMS scores. Any compilation failure counts as zero EMS in the mean win rate calculation. We note that GPT-4o, Claude 3.5 Sonnet, and Gemini 1.5 Pro are released after we have collected the data.

Table 2: Image2Struct evaluation results. The EMS score is conditioned on successful rendering. VLMs generally perform the best on Webpages, followed by LaTeX and then Musical Scores. The best performing model (GPT-4o) achieves a maximum rendering success rate of 0.977 and EMS of 0.708 on the 'easiest' domain (Webpages), indicating that the benchmark is not saturated.

| Model | Mean win rate | Image2Latex | | Image2Webpage | | Image2Music | |
|---|---|---|---|---|---|---|---|
| | | Rendering success | EMS | Rendering success | EMS | Rendering success | EMS |
| GPT-4 Omni | **0.923** | 0.807 | **0.660** | **0.980** | 0.710 | 0.491 | 0.340 |
| Claude 3.5 Sonnet | 0.821 | 0.756 | 0.611 | 0.972 | **0.724** | 0.455 | 0.317 |
| Gemini 1.5 Pro | 0.769 | 0.770 | 0.616 | 0.971 | 0.683 | 0.311 | 0.220 |
| Claude3 Opus | 0.731 | **0.836** | 0.632 | 0.655 | 0.378 | 0.551 | 0.367 |
| Gemini 1.0 Pro Vision | 0.615 | 0.519 | 0.402 | 0.789 | 0.499 | **0.583** | **0.402** |
| GPT-4 Vision | 0.577 | 0.758 | 0.598 | 0.957 | 0.653 | 0.000 | 0.000 |
| Palmyra Vision 003 | 0.564 | 0.738 | 0.537 | 0.884 | 0.640 | 0.103 | 0.072 |
| Claude3 Sonnet | 0.551 | 0.693 | 0.520 | 0.956 | 0.642 | 0.238 | 0.167 |
| LLaVA 1.6 (13B) | 0.385 | 0.345 | 0.247 | 0.731 | 0.447 | 0.417 | 0.258 |
| LLaVA 1.5 (13B) | 0.321 | 0.393 | 0.256 | 0.773 | 0.435 | 0.040 | 0.026 |
| Qwen-VL Chat | 0.256 | 0.732 | 0.525 | 0.031 | 0.008 | 0.000 | 0.000 |
| IDEFICS-instruct (9B) | 0.192 | 0.704 | 0.490 | 0.001 | 0.001 | 0.000 | 0.000 |
| IDEFICS 2 (8B) | 0.179 | 0.416 | 0.287 | 0.000 | 0.000 | 0.010 | 0.007 |
| IDEFICS-instruct (80B) | 0.115 | 0.357 | 0.240 | 0.001 | 0.001 | 0.000 | 0.000 |

## 4 Results

### 4.1 Model performance

Table 2 summarizes the performances of the VLMs and the breakdown across the different tasks and subtasks can be found in Appendix H. In general, we find that VLMs are unable to perform well on our tasks, with the best performing model achieving a maximum EMS of 0.724 in Webpages. Across all the models and subtasks, the average EMS is 0.324 for LaTeX, 0.370 for Webpages, and 0.069 for musical scores, indicating that the benchmark is not saturated and that there is a lot of room for VLMs to improve.

The VLMs perform better on Webpages and LaTeX than Musical Scores. For example, the best overall performing model (GPT-4o) achieves a rendering success rate (RSR) of 0.807 and an EMS score of 0.660 for LaTeX, an RSR of 0.980 and EMS of 0.710 for Webpages, but a RSR of only 0.491 and an EMS of only 0.340 for Musical Scores. We hypothesize that the disparity in performance between the domains may be due to the relative abundance of training data points describing LaTeX, HTML, CSS, and other formats used in web development in contrast to LilyPond. We observe differences within the domains too; for example, most models perform well on equations in LaTeX but struggle on plots (see Table A2).

Our initial benchmarking also shows that closed-API models perform significantly better than open-weight ones. The best-performing open-weight model has a lower mean win rate than the worst closed-API model. The top-performing models have different niches, and there is no model that outperforms the rest in all the tasks. While GPT-4o claims the overall top spot on our leaderboard, by being the overall best in Webpages and LaTeX while ranking third in Musical Scores. We reproduce some model predictions in Figure 4 and Appendix F.

We perform an error analysis (see Appendix I) and notice that the VLMs can extract the elements (e.g., text, images) but are unable to pick up on visual nuances. Furthermore, while some models are able to produce valid LilyPond code, none of the models we tested have the capability to interpret sheet music (see Figure 4c).

### 4.2 Comparison of metrics

We have introduced many image metrics to compare the rendered images against the input. In the case where ground-truth structures are available, we can compare the structures instead. Computing the structure similarity (e.g., using Levenshtein edit distance) potentially captures semantics to a

| | Reference | Claude 3 Opus | Gemini 1.5 Pro | GPT-4o | IDEFICS2 (8B) |
|---|---|---|---|---|---|
| | $\frac{S}{k_B} \leq \frac{4\pi r_s\,R}{4\ell_{Pl}^2}$ | $\frac{S}{k_B} \leq \frac{4\pi r_S^2 R}{4\ell_{Pl}^2}$ | $\frac{\mathcal{S}}{k_B} \geq \frac{4\pi r_s\,R}{4\ell_P^2\,l}$ | $\frac{S}{k_B} \leq \frac{4\pi r_s R}{4\ell_{Pl}^2}$ | 4. |
| Pixel Similarity | – | 0.000 | 0.000 | 0.000 | 0.000 |
| CIS | – | 0.990 | 0.980 | 0.986 | 0.856 |
| Earth Mover Similarity | – | 0.573 | 0.782 | 0.810 | 0.338 |
| Edit similarity (Levenshtein) | – | 0.320 | 0.400 | 0.462 | 0.000 |

(a) LaTeX (Equation) task. GPT-4o performed the best when it recreated the equation except for a single space between $R$ and the rest of the numerator. Gemini 1.5 Pro correctly inserted the space, but wrongly replaced the "$\leq$" with a "$\geq$" in addition to misinterpreting the subscript '$l$' in the denominator. Claude 3 Opus added a square term and mistakenly capitalized $s$. IDEFICS2 wasn't able to interpret the equation.

| | Reference | Claude 3 Opus | Claude 3 Sonnet | Gemini 1.5 Pro | GPT-4o | IDEFICS2 (8B) |
|---|---|---|---|---|---|---|
| Pixel Similarity | – | 0.079 | 0.041 | 0.079 | 0.079 | 0.000 |
| CIS | – | 0.784 | 0.766 | 0.815 | 0.752 | 0.819 |
| Earth Mover Similarity | – | 0.696 | 0.689 | 0.607 | 0.646 | 0.401 |
| Edit similarity (Levenshtein) | – | 0.239 | 0.270 | 0.261 | 0.293 | 0.000 |

(b) LaTeX (Plot) task. None of the models performed well, indicating this is a difficult instance.

| | | Pixel Similarity | CIS | Earth Mover Similarity |
|---|---|---|---|---|
| Reference | | – | – | – |
| Claude 3 Opus | | 0.000 | 0.628 | 0.626 |
| Gemini 1.5 Pro | | 0.000 | 0.600 | 0.587 |
| GPT-4o | | 0.000 | 0.600 | 0.599 |

(c) Music task. None of the models perform well on any instance in this domain. Ground truths are not available for this task.

Figure 4: Example model predictions for LaTeX (equation), LaTeX (Plot), and Music tasks. Results in the domain of Webpages, as well as additional instances of LaTeX, can be found in Appendix F.

better extent. We show the correlation between the various metrics in Table A1 and observe that the Earth Mover's Similarity (EMS) and cosine similarity between Inception vectors (CIS) correlate highly (0.79 and 0.80, respectively) with structural similarity.

## 4.3 Test on misspecified data

We test the models on 4 additional examples, 2 in LaTeX and 2 in Webpages. The input images for LaTeX are sourced from Word documents whereas those for Webpages are obtained from magazines (see Appendix J). The most powerful models, such as GPT-4o or Gemini 1.5 Pro, are able to produce valid structures despite the fact that images are generated from HTML or LaTeX. For LaTeX, the best rendered images contain the correct text and equations but cannot replicate the alignment of paragraphs. In some cases, the VLMs do not follow instructions to replicate the input but instead attempt to answer the questions in the images. For Webpages, the VLMs are able to extract most of the text but make mistakes in their positions or colors. This exercise demonstrates that our tasks are generalizable to structures that are not specified in the original language. Detailed results are available in Appendix K.

# 5   Related works

**VLM benchmarks.**   Benchmarks have been proposed to assess the ability of VLMs to perform different tasks, such as image captioning, visual question answering, or action recognition, among others [40, 20, 42, 43, 10, 16, 8]. However, existing benchmarks suffer from several limitations, such as ambiguity in evaluation, difficulty in scaling, and data leakage [4]. Benchmarks evaluate VLM responses either with human feedback [41, 40] or against reference outputs [20, 42, 43]. The former is suitable for the evaluation of long-form generations, but is expensive and difficult to reproduce. While some attempts use VLMs such as GPT-4V to evaluate the outputs of another, they are not yet reliable replacements for human annotators [45, 30]. Evaluation against reference outputs, on the other hand, are cheap and easily reproducible, but it is difficult to handle complex generation tasks. Furthermore, generating new test instances is non-trivial, and many benchmarks rely on existing data sources such as textbooks, prep books [43], or simply other databases [39, 30]. As a result, benchmarks are expensive to update and risk being incorporated into VLM training data, leading to data leakage [4]. In contrast, Image2Struct captures real-world use cases, is fully automatic and does not require human judgment, and uses fresh data so that it is difficult to game.

**Other benchmarks for image-to-code.**   Several datasets suitable for evaluating images-to-code have been released, some during Image2Struct's review period. Most of them focus on the task of converting images of webpages to HTML and contain data that are either manually curated or synthetic. Soselia et al. [32] creates two synthetic datasets with over 75,000 unique (code, screenshot) pairs to train a machine-learning model to produce HTML/CSS code from UI screenshots. Si et al. [30] curate a benchmark of 484 diverse real-world webpages by filtering the C4 dataset to test the ability of VLMs on converting visual designs into code implementations. Wan et al. [35] manually curate a dataset of "1,000 top-ranked real-world websites" and propose a method to translate webpage design to code. In contrast to Image2Struct, where the screenshots of the webpages are fed to the VLMs as-is, Wan et al. replaces images with placeholders and removes all external links and "elements that do not impact website's appearance" before rendering them as input images. Plot2Code [39] manually curates 132 matplotlib plots sourced from matplotlib galleries and tests VLMs in generating Python code from them. In contrast to these benchmarks, Image2Struct automatically obtains fresh, real-world data from online communities to test structure extraction from images across multiple domains.

**Metrics.**   There is a body of research on image comparison. Apart the metrics used in Image2Struct (i.e., LPIPS [44], EMD[26], or SSIM [36]), one may also employ the CLIP Score [30] or cross-correlation, or CIEDE2000 color difference[22]. The task of quantifying image similarity is related to pattern matching, where Siamese networks are used for facial recognition [5] and SIFT is used for local feature matching [21]. If the images are first broken down into visual elements (aka blocks), Block-match—which measures the ratio of matched block sizes to all block sizes—and the mean distance between matched blocks can be used to assess similarity [30, 35].

**Round-trip evaluation.**   The round-trip evaluation idea used in Image2Struct (input image $\rightarrow$ structure $\rightarrow$ rendered image) is related to backtranslation [28, 11, 12] and cycle-consistency [46, 9]. While existing works focus on *training* models for bidirectional mapping, we focus on *evaluating* the models.

# 6   Discussion

**VLM sensitivity to prompts and adaptations.**   VLMs are sensitive to prompts and our standardized text prompts may impact model evaluations. This is most evident when some models initially refuse to produce LilyPond codes but then comply when we state that the images are created by us. Beyond zero-shot prompting, there exist a variety of adaptation methods—specific procedures for invoking a model—such as chain-of-thoughts [37] or auto-prompt [29] that can enhance the performance of the models. We leave measuring the performance of VLMs under other adaptations as future work.

**Fine-tuning for structure extraction.**   Image2Struct proposes the task of extracting structure from images and uses a round-trip comparison methodology to evaluate the model. On the one hand, it would be interesting to explore fine-tuning VLMs using the round-trip supervision to perform these

useful tasks. On the other hand, developers may simply fine-tune on the Image2Struct dataset and therefore overfit on the tasks. Ideally, VLMs will acquire the capability to extract and reason over any structured data on image media.

**Updates with fresh data.** Image2Struct is amenable to rapid updates; the active online communities from which we download test instances from provide ample fresh data and the round-trip evaluation methodology avoids the need to label the newly downloaded data. As such we can quickly test new models with unseen data and thereby avoid the issue of data leakage. We note that the benchmark must be rerun on all the models after a data refresh so that an apples-to-apples comparison can be made. An exponential weighing scheme that prioritizes the latest batch of test instances can be used to incorporate both past and new test data.

## 6.1 Limitations

**Imperfect metrics.** The automated evaluation in Image2Struct hinges on having good metrics to compare the output image to the input image. The metrics introduced in this paper are not perfect, even though they have a high correlation with the edit distance between the generated and ground-truth source code. The EMS, for example, is still unable to discern if key elements exist in two images if the elements undergo other affine transformations beyond translation. Our work motivates future research in evaluating the similarity between rendered and ground-truth images, including developing evaluator VLMs that can be deployed as automatic evaluators to quantify image similarities.

**Limited scope of evaluation.** Our benchmark is limited in what it evaluates in several aspects. Firstly, it focuses solely on measuring structure extraction and does not capture tasks that involve either producing more abstract concepts or reasoning over extracted structure. Secondly, performing well on Image2Structure requires the VLMs to have knowledge of formal languages (e.g., HTML or LaTeX) in addition to visual understanding; as such, it does not discern between a model that is excellent at understanding structured data but is poor in a language, and, an model that is simply poor at understanding structured information. Thirdly, we do not measure robustness of the VLMs to noise (e.g., scans) and other perturbations (e.g., image skew). We leave these possible extensions as future work.

## 6.2 Broader Impacts

The task of extracting the structured data embedded in an input is applicable to a very broad and practical set of tasks. Beyond parsing images of webpages, scientific documents, and sheet music, we envision the same framework can be extended to extract information from visual content in various domains (e.g., radiology images, electronic health records) and modalities (e.g., 3D graphics). It is possible that in the future, one will be able to command a VLM to convert a natural image into a structured form so that it can be edited.

This work also introduces automatic, repeatable, and reproducible evaluation methods, as well as method to produce fresh test sets for fair evaluation. This promotes transparency and accountability in model development and stakeholders (including researchers, developers, and policymakers) can better understand and compare the performance of different VLMs.

Our evaluation framework allows the development of powerful VLMs which may displace existing workers (e.g., data entry personnel). The ease of replicating and editing rendered structures can be misused and we urge developers to consider the implications of their technology and take appropriate measures to safeguard against misuse.

## 7 Conclusion

We introduced Image2Struct, a benchmark for evaluating VLMs in extracting structure from images, such as webpages, LaTeX, and music sheets. We enabled evaluation on fresh and diverse test examples by continuously downloading real, latest data, and developing automated metrics that compare images rendered from model predictions against the original images.

## Acknowledgements

We thank Google for their support for the project. The views and opinions expressed in this article are those of the authors only and do not necessarily represent the views and opinions of any other organization, any of their affiliates or employees acknowledged above.

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

**Appendix**

## A    Classical metrics

**Pixel Similarity.**    The pixel similarity (PS) is defined as the average of the binary comparisons between pixels in two images.

$$\text{PS}(x, \hat{x}) = \frac{1}{WH} \sum_{i \in \{1, \cdots, W\}} \sum_{j \in \{1, \cdots, H\}} \mathbb{1}\{x(i, j) = \hat{x}(i, j)\} \tag{5}$$

While it captures the idea that $\hat{x}$ should be identical to $x$, it leaves little room for error and may generate a poor score even when $\hat{x}$ is simply a pixel shift of $x$. This may cause the benchmark to suffer from a lack of ability to differentiate the models' performances and produce false rankings. Our implementation ignores the modal color (usually the background) in the union of $x$ and $\hat{x}$ and allows some minor color differences (i.e., pixels in the same location are considered the same if their color values are within 2% of one another).

**SSIM.**    SSIM [36] is a score that compares the luminance, contrast and pixel variance between two images. We use the implementation provided by `scikit-image` and refer readers to Zhou et al. [36] for calculation details. We normalize the score to be between 0 and 1 and modify it such that a higher value indicates higher similarity (i.e., 1 indicates exact match).

**LPIPS.**    LPIPS [44] has been shown to match human perception and is similar to CIS in that it applies a distance metric between the activations of a neural network. We use the implementation provided by `torchvision` and choose VGG as the neural network of interest. We refer readers to the original paper for implementation details. We normalize the scores to be between 0 and 1, where a higher score means that the images are more similar.

**Earth Mover Distance**    To compute the classic EMD between two images $x$ and $\hat{x}$, we first transform the images into signatures $\mathcal{S}(x)$ and $\mathcal{S}(\hat{x})$, which are discrete distributions of features of $Q$ elements. The signature is defined as the distribution of the grayscale values of an image when one wants to compare images. In other words, $\mathcal{S}(x)$ is the probability mass function where the random variable (i.e., $g_k$ or $h_l$) is one of the possible pixel values (0 to 255) and the mass (i.e., $w_k^g$ or $w_l^h$) is the normalized count of the number of pixels in $x$ with that value.

$$\mathcal{S}(x) = \{(g_k, w_k^g) : 1 \leq k \leq Q\} \qquad \& \qquad \mathcal{S}(\hat{x}) = \{(h_l, w_l^h) : 1 \leq l \leq Q\} \tag{6}$$

We then define a cost matrix $C \in \mathbb{R}^{Q \times Q}$ wherein each element $C_{k,l}$ represents the cost of moving probability mass between $g_k$ and $h_l$. We further denote the movement of probability mass between $g_k$ and $h_l$ by $f_{k,l}$. The optimal flow is the set of $\{f_{k,l}^*\}$

that satisfies the following optimization problem:

$$\min \sum_k \sum_l f_{k,l} C[k,l] \qquad \text{subject to} \qquad (7a)$$

$$f_{k,l} \geq 0 \qquad \forall k, \forall l \qquad (7b)$$

$$\sum_l f_{k,l} \leq w_k^g \qquad \forall k \qquad (7c)$$

$$\sum_k f_{k,l} \leq w_k^h \qquad \forall l \qquad (7d)$$

$$\sum_l \sum_k f_{k,l} = \min\{\sum_k w_k^g, \sum_l w_l^h\} \qquad (7e)$$

The EMD can then be computed with Equation (8).

$$\text{EMD}(x, \hat{x}) = \frac{\sum_k \sum_l f_{k,l}^* C[k,l]}{\sum_k \sum_l f_{k,l}^*} \qquad (8)$$

## B Illustration of EMD$_{\text{block}}$

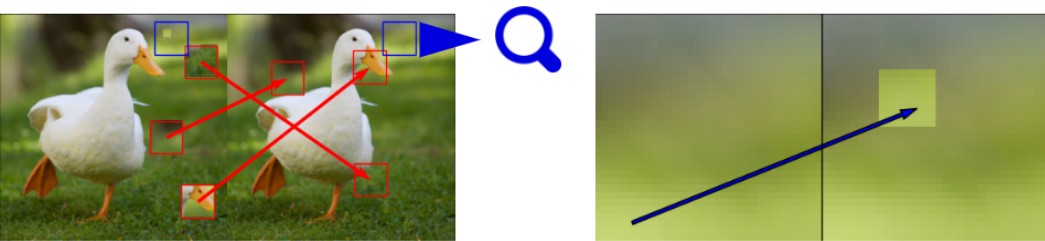

Figure A1: An illustration of the two scales at which EMD$_{\text{block}}$ operates. The left image is an altered copy of the right one in that 4 patches are manipulated. EMD$_{\text{block}}$ computes an optimal flow where 3 of these patches (in red) are moved completely without modification. For the blue patch, it decides that it incurs a lower cost to move some pixels within the patch (the zoomed version on the right). On top of moving blocks or pixels, EMD$_{\text{block}}$ can change the pixel colors at a cost (we do not illustrate color modification in this example for simplicity).

## C  Prompts

We detail the text prompts provided to the VLMs in this section.

### Music

Just give a short answer without answering in a complete sentence.
Please generate the Lilypond code to generate a music sheet that looks like this image as much as feasibly possible. This music sheet was created by me, and I would like to recreate it using Lilypond.

### LaTeX

Please provide the LaTeX code used to generate this image. Only generate the code relevant to what you see. Your code will be surrounded by all the imports necessary as well as the begin and end document delimiters.

### Webpages

Please generate the source code to generate a webpage that looks like this image as much as feasibly possible. You should output a JSON object associating each file name with its content.
Here is a simple example of the expected structure (that does not correspond to the image). In this example, 3 files are created: index.html, style.css and script.js.

```
[
  {
    "filename": "index.html",
    "content": "<!DOCTYPE html>\n<html>\n<head>\n<title> Title of the document
                </title>\n</head>\n<body>\n\n<p>Content of the document......</p>\n\n </body>\n</html>"
  },
  {
    "filename": "style.css",
    "content": "body \n background-color: lightblue;\n\nh1 \n color: white;\n text-align: center;\n"
  },
  {
    "filename": "script.js",
    "content": "document.getElementById(\"demo\").innerHTML = \"Hello JavaScript!\";"
  }
```

] You do not have to create files with the same names. Create as many files as you need, you can even use directories if necessary, they will be created for you automatically. Try to write some realistic code keeping in mind that it should look like the image as much as feasibly possible.

# D Post-processing

A few post-processing steps are taken to maximize compilation success.

Across all domains, we first scan for headers and remove them. This is necessary because some VLMs wrap code within header blocks. For example, Gemini 1.5 Pro wraps the actual code with "'[code]'" whereas LLaVa v1.6 and GPT-4V wrap them with "'`latex` [code]'".

For LaTeX documents, we then do the following:

1. Remove all the preambles (if any) from the code.
2. Add preambles (e.g., `\begin{document}` and `\end{document}`), custom imports and document style. This and the previous step are implemented to standardize the rendering and to maximize compilation success by avoiding conflicting packages in the compilation environment.
3. Attempt to fix "`Missing $`", "`math mode missing`", or similar errors by adding `\begin{equation}` and `\end{equation}` to the code block.
4. Attempt to import missing packages; this is detected when `LaTeX Error: Environment (.*) undefined` is raised during compilation. When this happens, we try to pass an import with the environment name, which will fix the issue if the package contains the required environment.

For Musical Scores, we do the following:

1. Invoke the `convert-ly` command in LilyPond to convert the VLM output to the current version of LilyPond to avoid issues where the output is valid for an older version of LilyPond.

For Webpages, no additional post-processing is done since the output should be a valid JSON file.

# E  Example of multiple correct answers

We present two examples of LaTeX code that compiles to the same equation. The edit similarity between them is 0.46.

```
\documentclass{article}
\usepackage{amsmath}
\begin{document}
\[
(a+b)^n = \sum_{k=0}^{n} \binom{n}{k} a^{n-k} b^k
\]
\end{document}
```

Figure A2: The first example LaTeX code that compiles to $(a + b)^n = \sum_{k=0}^{n} \binom{n}{k} a^{n-k} b^k$.

```
\documentclass{article}
\usepackage{amsmath}
\newcommand{\bsum}{\sum}
\newcommand{\bcoef}[2]{\binom{#1}{#2}}
\newcommand{\apower}[2]{#1^{#2}}
\begin{document}
\[
(a+b)^n = \bsum_{k=0}^{n} \bcoef{n}{k} \apower{a}{n-k} \apower{b}{k}
\]
\end{document}
```

Figure A3: The second example LaTeX code that also compiles to $(a + b)^n = \sum_{k=0}^{n} \binom{n}{k} a^{n-k} b^k$.

# F Examples

## F.1 LaTeX – Algorithm

**Reference**

```
1: procedure MODIFIEDGD(∇F, L̂_{i,0}, δ_{(i,·)}, R, θ_{i,0}, κ_{(i,·)}, T*, ℓ, γ)
2:     Δ ← 0                                    ▷ Measure of predicted decrease of objective
3:     for j = 0, 1, ..., T* − 1 do
4:         α_{i,j} ← δ_{(i,·)}(2||∇F(θ_{i,j})||³ + ||∇F(θ_{i,j})||²L̂_{i,j})⁻¹        ▷ Step size
       computation
5:         θ_{i,j+1} ← θ_{i,j} − α_{i,j}∇F(θ_{i,j})                          ▷ Gradient step
6:         Δ ← Δ − δ²_{(i,·)}(4||∇F(θ_{i,j})||³ + 2L̂_{i,j}||∇F(θ_{i,j})||²)⁻¹   ▷ Update
       predicted decrease
7:         if ||θ_{i,j} − θ_{i,j+1}|| ≥ 10⁻¹⁰ then
8:             L̂_{i,j+1} ← UPDATEL(∇F, θ_{i,j}, θ_{i,j+1}, L̂_{i,j}, κ_{(i,·)}, ℓ)   ▷ Update local
       Lipschitz estimate
9:         else
10:            L̂_{i,j+1} ← L̂_{i,j}
11:        end if
12:        δ_{cond} ← ||∇F(θ_{i,j+1})||² ∉ (δ_{(i,·)}, γδ_{(i,·)})
13:        if δ_{cond} or ||θ_{i,j+1} − θ_{i,0}|| > R or stop condition then        ▷ Check
       triggering events
14:            return θ_{i,j+1}, L̂_{i,j+1}, δ_{cond}, Δ
15:        end if
16:    end for
17:    return θ_{i,T*}, L̂_{i,T*}, δ_{cond}, Δ
18: end procedure
```

**Prediction (GPT-4o)**

```
1: procedure MODIFIEDGD(∇F, L̂_{i,0}, R, δ(i,.), θ_{i,0}, κ(i,.), T*, ℓ, γ)
2:     Δ ← 0                                    ▷ Measure of predicted decrease of objective
3:     for j = 0, 1, ..., T* − 1 do
4:         α_{i,j} ← δ(i,.)(2||∇F(θ_{i,j})||³ + ||∇F(θ_{i,j})||²L̂_{i,j})⁻¹        ▷ Step size
       computation
5:         θ_{i,j+1} ← θ_{i,j} − α_{i,j}∇F(θ_{i,j})                          ▷ Gradient step
6:         Δ ← Δ − δ²(i,.)(4||∇F(θ_{i,j})||³ + 2L̂_{i,j}||∇F(θ_{i,j})||²)⁻¹   ▷ Update
       predicted decrease
7:         if ||θ_{i,j} − θ_{i,j+1}|| ≥ 10⁻¹⁰ then
8:             L̂_{i,j+1} ← UpdateL(∇F, θ_{i,j}, θ_{i,j+1}, L̂_{i,j}, κ(i,.), ℓ)   ▷ Update local
       Lipschitz estimate
9:         else
10:            L̂_{i,j+1} ← L̂_{i,j}
11:        end if
12:        δ_{cond} ← ||∇F(θ_{i,j+1})||² ∉ (δ(i,.), γδ(i,.))
13:        if δ_{cond} or ||θ_{i,j+1} − θ_{i,j}|| > R or stop condition then        ▷ Check
       triggering events
14:            return θ_{i,j+1}, L̂_{i,j+1}, δ_{cond}, Δ
15:        end if
16:    end for
17:    return θ_{i,T*}, L̂_{i,T*}, δ_{cond}, Δ
18: end procedure
```

| Pixel Similarity | CIS | Earth Mover Similarity | Edit similarity (Levenshtein) |
|---|---|---|---|
| 0.561 | 0.991 | 0.962 | 0.789 |

Figure A4: Example of a good successful compilation on a LaTeX (Algorithm) task. Note the mistake where "\kappa_(i,\cdot)" $\kappa_{(i,\cdot)}$ is substituted with "\kappa(i,.)" $\kappa(i,.)$. VLMs still have trouble perceiving subtle differences in font and shape sizes.

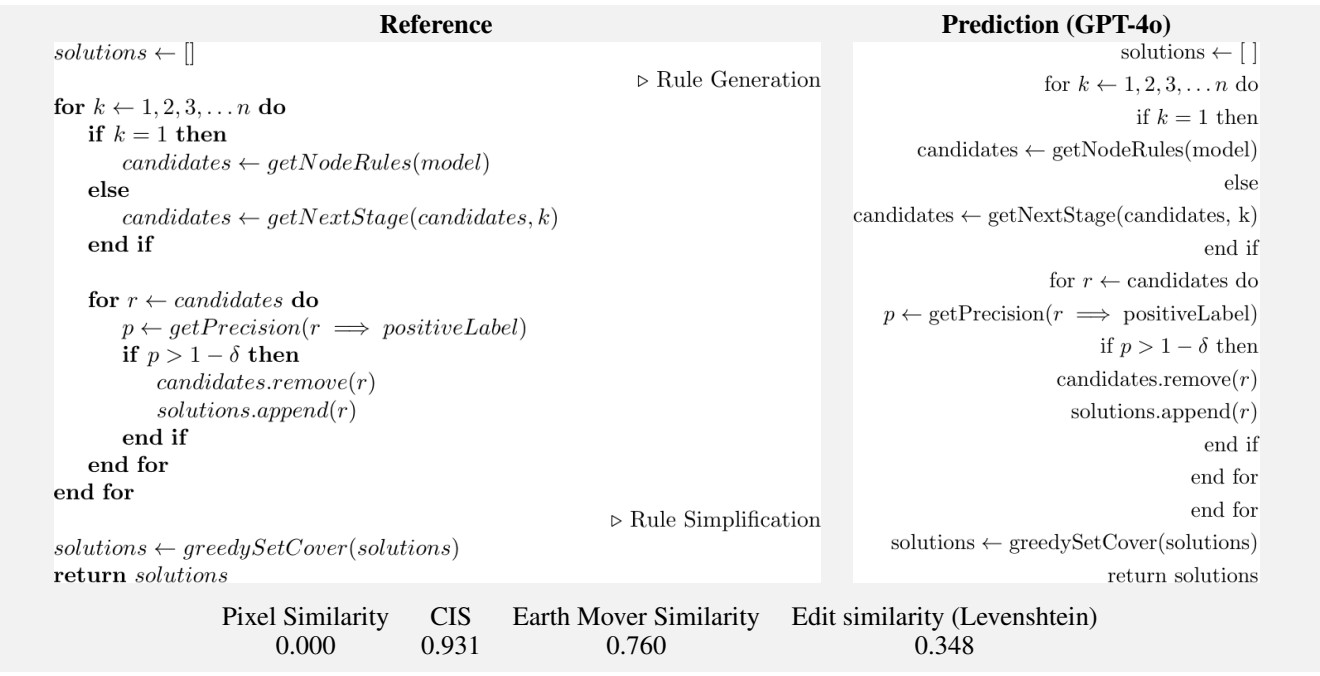

Figure A5: Example of a moderately successful compilation on a LaTeX (Algorithm) task. The VLM is able to recognize the text in the input image but does not format the output as an algorithm. Comments are also ignored.

| Reference | Prediction (IDEFICS2-8B) |
|---|---|

**Algorithm 1** Independent Metropolis-Hastings (IMH) Algorithm

Pick an initial state $(x_0, y_0, z_0) \sim \nu(x, y, z)$.
**for** $i = 0, 1, 2, \ldots$ **do**
    Generate a random candidate state $(x', y', z') \sim \nu(x, y, z)$.
    Calculate the acceptance probability $A(x_i, y_i, z_i, x', y', z') =$
$\min \left\{ 1, \frac{\pi(x', y', z')\nu(x_i, y_i, z_i)}{\pi(x_i, y_i, z_i)\nu(x', y', z')} \right\}$
    Generate a uniform random number $u \in [0, 1]$.
    **if** $u \leq A(x_i, y_i, z_i, x', y', z')$ **then**
        accept the new state $(x', y', z')$ and set $(x_{i+1}, y_{i+1}, z_{i+1}) = (x', y', z')$.
    **else**
        reject the new state $(x', y', z')$ and set $(x_{i+1}, y_{i+1}, z_{i+1}) = (x_i, y_i, z_i)$.
    **end if**
**end for**

Prediction text:

1 Independent Metropolis-Hastings (IMH) Algorithm Pick an initial state (x0,y0,z0) for = 0.1, 2, ... do Generate a random candidate state (x*,y*,z*) Calculate the acceptance probability A(x*,y*,z—x0,y0,z0) Generate a uniform random number u [0,1] If u ¡ A(x*,y*,z—x0,y0,z0) then accept the new state (x*,y*,z) and set (x1,y1,z1) = (x*,y*,z) else reject the new state (x*,y*,z) and set (x1,y1,z1) = (x0,y0,z0) end if end for

| Pixel Similarity | CIS | Earth Mover Similarity | Edit similarity (Levenshtein) |
|---|---|---|---|
| 0.000 | 0.925 | 0.759 | 0.129 |

Figure A6: Example of a poor successful compilation on a LaTeX (Algorithm) task. The VLM is able to recognize some text in the input image but does not format the output as an algorithm. Comments are also ignored.

## F.2   Webpages – HTML

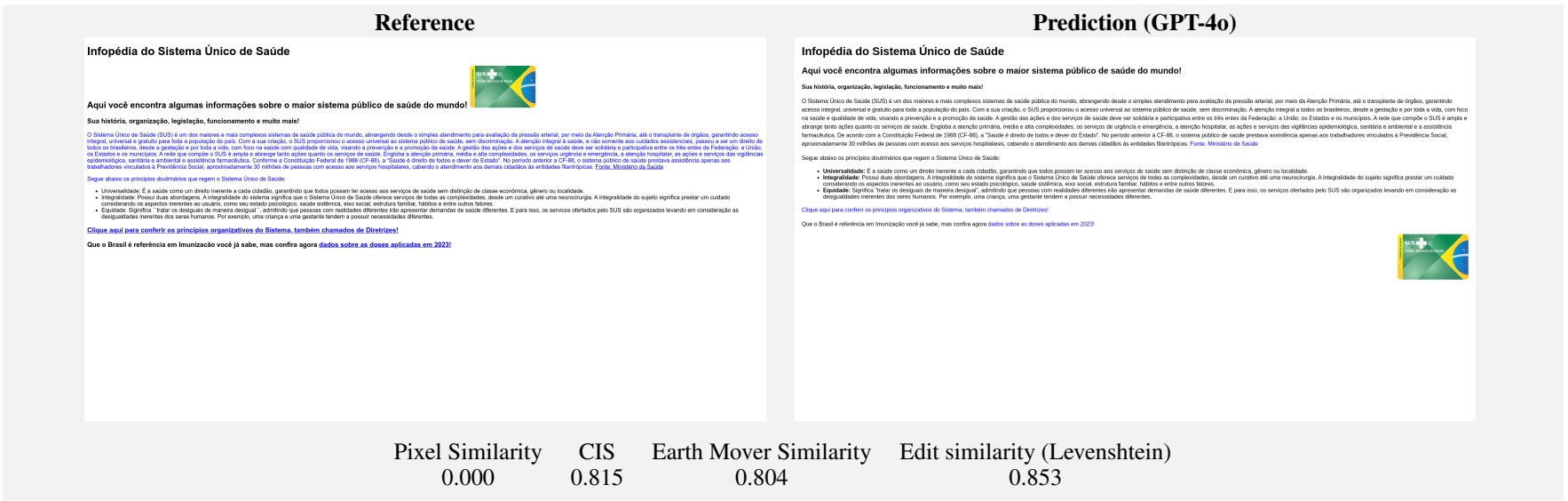

| Pixel Similarity | CIS | Earth Mover Similarity | Edit similarity (Levenshtein) |
|:---:|:---:|:---:|:---:|
| 0.000 | 0.815 | 0.804 | 0.853 |

Figure A7: Example of a good successful reproduction for Webpages (HTML). The text is correct but the colors of the paragraphs do not match. Interestingly, the VLM added bold font for the first word in the bullet points. The line spacing and image position are not reproduced faithfully.

## G  Correlation between metrics

Table A1: Correlation between the different metrics across all the subdomains. Only examples where the predicted structures render successfully are used when computing correlation. Edit similarity is computed on structures and is only available for Webpages and LaTeX.

|  | EMS | Pixel Similarity | CIS | Edit Similarity | SSIM | LPIPS |
|---|---|---|---|---|---|---|
| EMS | 1.000 | 0.350 | 0.979 | 0.794 | 0.934 | 0.738 |
| Pixel Similarity | 0.350 | 1.000 | 0.324 | 0.324 | 0.169 | -0.179 |
| CIS | 0.979 | 0.324 | 1.000 | 0.803 | 0.936 | 0.768 |
| Edit Similarity | 0.794 | 0.324 | 0.803 | 1.000 | 0.783 | 0.489 |
| SSIM | 0.934 | 0.169 | 0.936 | 0.783 | 1.000 | 0.825 |
| LPIPS | 0.738 | -0.179 | 0.768 | 0.489 | 0.825 | 1.000 |

## H  Detailed results

We report detailed statistics from our initial benchmarking in this section.

Table A2: Average performances of VLMs on the LaTeX task. The scores reported are conditioned on successful rendering.

| Model | Compilation success | EMS | Pixel Similarity | CIS | LPIPS | SSIM | Edit sim. |
|---|---|---|---|---|---|---|---|
| GPT-4o (2024-05-13) | 0.807 | **0.660** | **0.146** | **0.745** | 0.198 | **0.568** | **0.420** |
| Claude 3 Opus (20240229) | 0.836 | 0.632 | 0.041 | 0.733 | 0.289 | 0.525 | 0.303 |
| Gemini 1.5 Pro (0409 preview) | 0.771 | 0.616 | 0.100 | 0.707 | 0.211 | 0.523 | 0.347 |
| Claude 3.5 Sonnet (20240620) | 0.756 | 0.611 | 0.113 | 0.693 | 0.192 | 0.529 | 0.382 |
| GPT-4V (1106 preview) | 0.758 | 0.598 | 0.078 | 0.684 | 0.214 | 0.500 | 0.332 |
| Palmyra Vision 003 | 0.738 | 0.537 | 0.039 | 0.617 | 0.274 | 0.488 | 0.177 |
| Qwen-VL Chat | 0.732 | 0.525 | 0.002 | 0.570 | **0.319** | 0.459 | 0.042 |
| Claude 3 Sonnet (20240229) | 0.693 | 0.520 | 0.030 | 0.596 | 0.237 | 0.456 | 0.253 |
| IDEFICS-instruct (9B) | 0.704 | 0.490 | 0.002 | 0.548 | 0.296 | 0.468 | 0.069 |
| Gemini 1.0 Pro Vision | 0.520 | 0.403 | 0.051 | 0.466 | 0.141 | 0.355 | 0.201 |
| IDEFICS 2 (8B) | 0.416 | 0.287 | 0.001 | 0.329 | 0.177 | 0.262 | 0.025 |
| LLaVA 1.5 (13B) | 0.393 | 0.256 | 0.002 | 0.327 | 0.152 | 0.254 | 0.095 |
| IDEFICS-instruct (80B) | 0.357 | 0.240 | 0.004 | 0.264 | 0.157 | 0.248 | 0.076 |
| LLaVA 1.6 (13B) | 0.345 | 0.247 | 0.004 | 0.300 | 0.128 | 0.215 | 0.097 |

Table A3: Average performances of VLMs on the Webpages task. The scores reported are conditioned on successful rendering.

| Model | Compilation success | EMS | Pixel Similarity | CIS | LPIPS | SSIM | Edit sim. |
|---|---|---|---|---|---|---|---|
| Claude 3.5 Sonnet (20240620) | 0.972 | **0.724** | **0.010** | **0.822** | 0.285 | **0.781** | 0.589 |
| GPT-4o (2024-05-13) | 0.980 | 0.710 | 0.009 | 0.819 | 0.333 | 0.771 | **0.590** |
| Gemini 1.5 Pro (0409 preview) | 0.971 | 0.683 | 0.010 | 0.798 | 0.367 | 0.738 | 0.556 |
| GPT-4V (1106 preview) | 0.957 | 0.653 | 0.009 | 0.774 | 0.378 | 0.708 | 0.512 |
| Claude 3 Sonnet (20240229) | 0.956 | 0.642 | 0.004 | 0.771 | **0.383** | 0.678 | 0.492 |
| Palmyra Vision 003 | 0.884 | 0.640 | 0.006 | 0.730 | 0.311 | 0.690 | 0.508 |
| Gemini 1.0 Pro Vision | 0.795 | 0.502 | 0.001 | 0.617 | 0.316 | 0.562 | 0.332 |
| LLaVA 1.5 (13B) | 0.773 | 0.435 | 0.001 | 0.570 | 0.379 | 0.530 | 0.073 |
| LLaVA 1.6 (13B) | 0.731 | 0.447 | 0.001 | 0.550 | 0.332 | 0.503 | 0.138 |
| Claude 3 Opus (20240229) | 0.655 | 0.378 | 0.002 | 0.504 | 0.319 | 0.423 | 0.203 |
| Qwen-VL Chat | 0.031 | 0.008 | 0.000 | 0.021 | 0.017 | 0.018 | 0.000 |
| IDEFICS-instruct (80B) | 0.002 | 0.001 | 0.000 | 0.001 | 0.001 | 0.001 | 0.000 |
| IDEFICS-instruct (9B) | 0.001 | 0.001 | 0.000 | 0.001 | 0.001 | 0.001 | 0.000 |
| IDEFICS 2 (8B) | 0.000 | 0.000 | 0.000 | 0.000 | 0.000 | 0.000 | 0.000 |

Table A4: Average performances of VLMs on the Music task. The scores reported are conditioned on successful rendering.

| Model | Compilation success | EMS | Pixel Similarity | CIS | LPIPS | SSIM | Edit Sim. |
|---|---|---|---|---|---|---|---|
| Gemini 1.0 Pro Vision | **0.583** | **0.402** | **0.000** | **0.411** | 0.284 | **0.332** | – |
| Claude 3 Opus (20240229) | 0.551 | 0.367 | 0.000 | 0.387 | **0.287** | 0.285 | – |
| GPT-4o (2024-05-13) | 0.491 | 0.340 | 0.000 | 0.350 | 0.250 | 0.262 | – |
| Claude 3.5 Sonnet (20240620) | 0.455 | 0.317 | 0.000 | 0.340 | 0.236 | 0.243 | – |
| LLaVA 1.6 (13B) | 0.417 | 0.258 | 0.000 | 0.279 | 0.209 | 0.247 | – |
| Gemini 1.5 Pro (0409 preview) | 0.311 | 0.220 | 0.000 | 0.222 | 0.151 | 0.181 | – |
| Claude 3 Sonnet (20240229) | 0.238 | 0.167 | 0.000 | 0.179 | 0.125 | 0.127 | – |
| Palmyra Vision 003 | 0.103 | 0.072 | 0.000 | 0.074 | 0.052 | 0.054 | – |
| LLaVA 1.5 (13B) | 0.040 | 0.026 | 0.000 | 0.029 | 0.019 | 0.023 | – |
| IDEFICS 2 (8B) | 0.010 | 0.007 | 0.000 | 0.007 | 0.005 | 0.006 | – |
| Qwen-VL Chat | 0.000 | 0.000 | 0.000 | 0.000 | 0.000 | 0.000 | – |
| IDEFICS-instruct (9B) | 0.000 | 0.000 | 0.000 | 0.000 | 0.000 | 0.000 | – |
| IDEFICS-instruct (80B) | 0.000 | 0.000 | 0.000 | 0.000 | 0.000 | 0.000 | – |
| GPT-4V (1106 preview) | 0.000 | 0.000 | 0.000 | 0.000 | 0.000 | 0.000 | – |

# I   Results of error analysis

For our error analysis of the state-of-the-art models, we randomly selected 7 examples for each subdomain (i.e., LaTeX–Algorithm, LaTeX–Equation, ...) and manually analyzed 56 compiled images for GPT-4o.

Across all domains, we observe that GPT-4o is capable of extracting text with minor (no change in meaning) or no error for 42 of the 48 instances (the remaining 7 instances of music do not contain relevant text).

## I.1   LaTeX

When we look at the 14 instances in LaTeX equations and algorithms, we find that the common mistakes are 1) failure to insert newlines (3 of 14), 2) failure to make symbols italics or bold font (3 of 14), and 3) using the wrong modifiers such as hats and bars (1 of 14).

$$\tilde{Y}_i^{(q)}(1) = D_i Y_i + (1 - D_i) Y_i^{m,(q)}$$
$$\tilde{Y}_i^{(q)}(0) = D_i Y_i^{m,(q)} + (1 - D_i) Y_i.$$

$$\tilde{Y}_i^{(q)}(1) = D_i Y_i + (1 - D_i) Y_i^{m,(q)} \tilde{Y}_i^{(q)}(0) = D_i Y_i^{m,(q)} + (1 - D_i) Y_i.$$

(a) Ground Truth                      (b) Prediction

Figure A8: Example of a prediction missing new lines

(a) Ground Truth                      (b) Prediction

Figure A9: Example of a prediction with incorrect bolded text

(a) Ground Truth                      (b) Prediction

Figure A10: Example of a prediction with incorrect modifiers (circled in purple)

For the 7 LaTeX plots, we observe while all text is extracted and simple geometric shapes (e.g., rectangles, diamonds, or lines) are created, none of the rendered images is satisfactory; GPT-4o attempts to replicate the color of the elements but only succeeds when the colors used are the primary ones (e.g., 'red' or 'blue' in `xcolor` but no colors defined with RGB values). The relative sizes and positions of the elements are not respected and the points in scatterplots seem to be randomly generated.

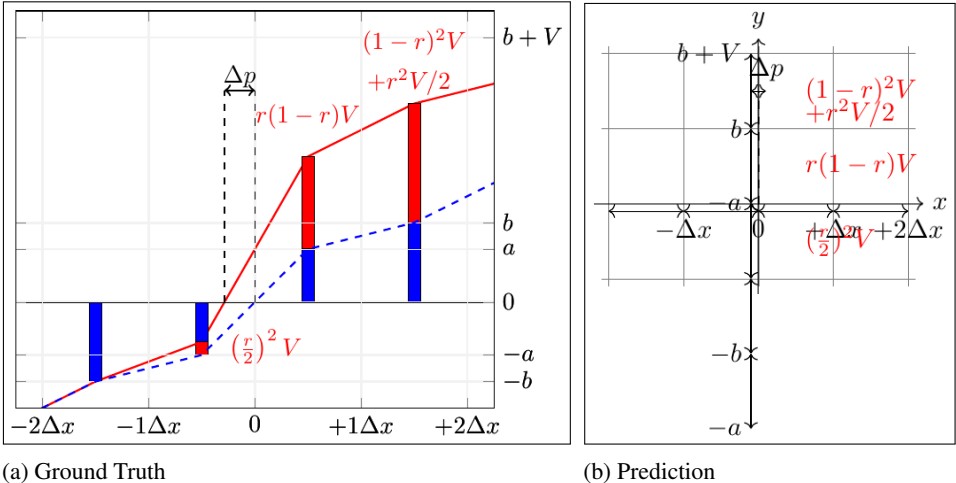

(a) Ground Truth

(b) Prediction

Figure A11: Example of a plot prediction that is not satisfactory

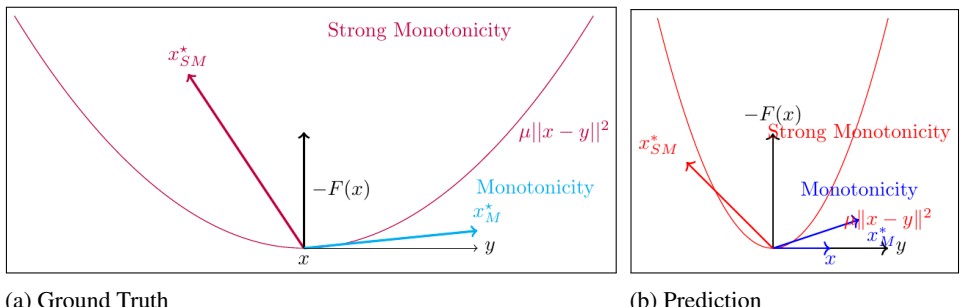

(a) Ground Truth

(b) Prediction

Figure A12: Example of a prediction with incorrect colors (the prediction is using default red and blue colors)

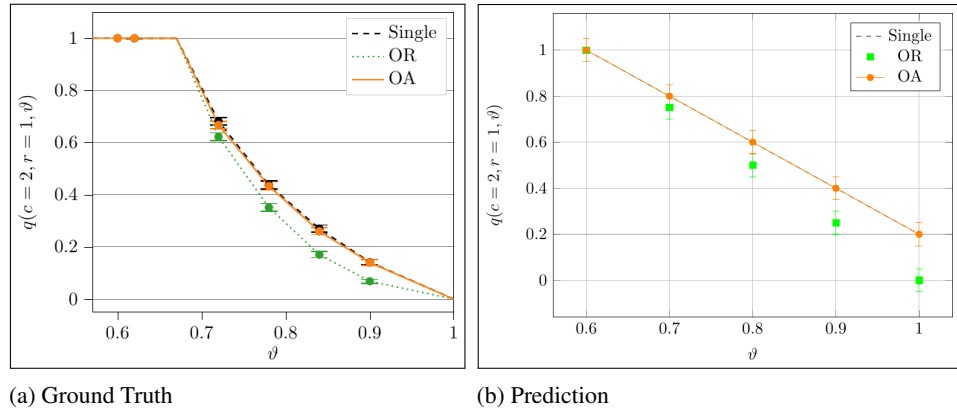

(a) Ground Truth

(b) Prediction

Figure A13: Example of a prediction with incorrect numerical values

GPT-4o is able to replicate LaTeX tables with minor or no mistakes 4/7 of the time. In 2 of the remaining 3 instances, it fails to parse the data correctly, resulting in wrong or missing data. In the final instance, it missed out a tilde above one of the symbols, which drastically changed the meaning.

Table 1: Noise Detection Results for Multiclass Synthetic Dataset under NCAR and NNAR noise types

| noise | method | NCAR FPR | NCAR FNR | NNAR FPR | NNAR FNR |
|---|---|---|---|---|---|
| 10% | thresh $m_j$ | 06.5 | 01.6 | 06.5 | 15.4 |
| | thresh $w_j$ | 06.7 | 01.8 | 06.0 | 18.2 |
| | cleanlab | 02.1 | 09.0 | 02.1 | *74.3* |
| | doubtlab | 07.3 | 22.8 | 06.5 | *59.7* |
| 20% | thresh $m_j$ | 06.5 | 02.1 | 01.9 | 65.5 |
| | thresh $w_j$ | 06.7 | 01.9 | 01.6 | 68.9 |
| | cleanlab | 03.4 | 06.4 | 01.3 | *82.2* |
| | doubtlab | 10.2 | 02.0 | 02.3 | 23.8 |
| 30% | thresh $m_j$ | 14.1 | 16.1 | 05.9 | *58.1* |
| | thresh $w_j$ | 09.8 | 13.4 | 02.9 | *65.3* |
| | cleanlab | 17.5 | 15.2 | 01.8 | *85.2* |
| | doubtlab | *73.8* | 00.2 | 02.7 | 19.7 |

(a) Ground Truth

Table 1: Noise Detection Results for Multiclass Synthetic Dataset under NCAR and NNAR noise types

| noise | method | NCAR FPR | NCAR FNR | NNAR FPR | NNAR FNR |
|---|---|---|---|---|---|
| 10% | thresh $m_j$ | 06.5 | 01.6 | 06.5 | 15.4 |
| | thresh $w_j$ | 06.7 | 01.8 | 06.0 | 18.2 |
| | cleanlab | 02.1 | 09.0 | 02.1 | 74.3 |
| | doubtlab | 07.3 | 22.8 | 06.5 | 59.7 |
| 20% | thresh $m_j$ | 06.5 | 02.1 | 01.9 | 65.5 |
| | thresh $w_j$ | 06.7 | 01.9 | 01.6 | 68.9 |
| | cleanlab | 03.4 | 06.4 | 01.3 | 82.2 |
| | doubtlab | 10.2 | 02.0 | 02.3 | 23.8 |
| 30% | thresh $m_j$ | 14.1 | 16.1 | 05.9 | 58.1 |
| | thresh $w_j$ | 09.8 | 13.4 | 02.9 | 65.3 |
| | cleanlab | 17.5 | 18.1 | 05.8 | 85.2 |
| | doubtlab | 17.8 | 02.7 | 02.7 | 19.7 |

(b) Prediction

Figure A14: Example of a predicted table with incorrect data (circled in purple)

## I.2 Musical scores

For Musical Scores, all 7 instances seem to be randomly generated and do not have the correct key, number of measures, or notes. However, this is not due to a lack of knowledge of the Lilypond syntax as GPT-4o get a compilation success of 0.491 on the entire dataset.

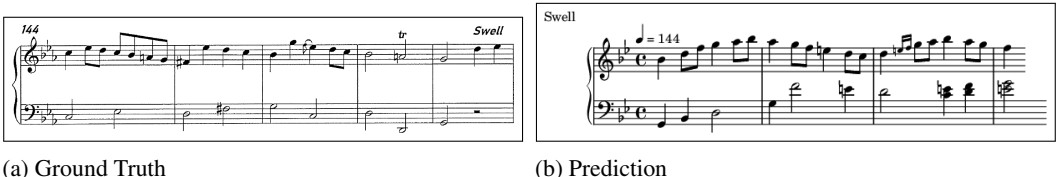

(a) Ground Truth

(b) Prediction

Figure A15: Example of a predicted musical scores that has nothing to do with the original one

## I.3 Webpages

For Webpages, GPT-4o manages to reproduce the text and simple elements in all the 21 instances analyzed. We notice that the model struggles with backgrounds with color gradients in all the 3 instances where they appear (similar issue as color errors in LaTeX plots). In 12 of the instances, we observe bad relative size, position, or alignment of the elements, resulting in low scores. The substance is captured (e.g., there is a "Login" button) but the form is very often incorrect (e.g., the button has a different font, color and is not placed where it should be)

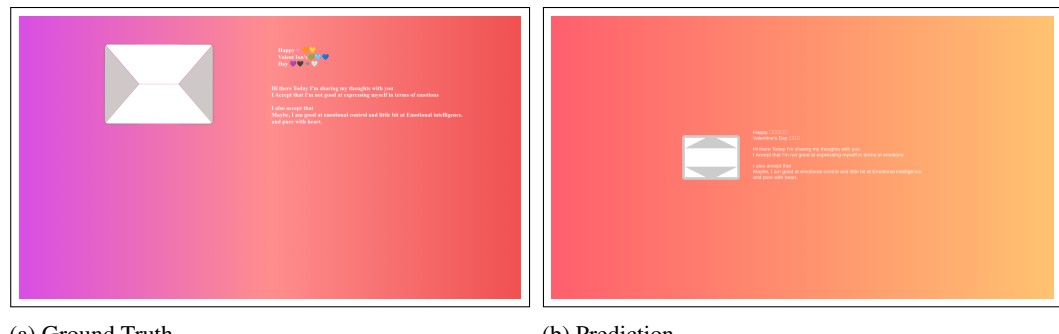

(a) Ground Truth         (b) Prediction

Figure A16: Example of a prediction with an incorrect color gradient (as well as a layout issue)

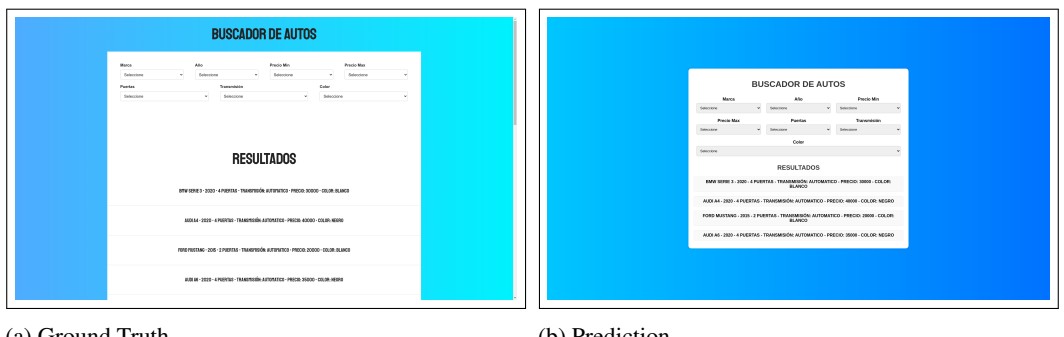

(a) Ground Truth         (b) Prediction

Figure A17: Example of a prediction with correct elements but with a different layout

## J Misspecified data

Which of the following statements refutes the claim that $GL_R(3)$, the set of $3 \times 3$ invertible matrices over the real numbers, is a field?

A.  There exist elements $A$ and $B$ of $GL_R(3)$ such that $AB \neq BA$.

B.  There exist elements $A$ and $B$ of $GL_R(3)$ such that $\det(AB) = \det(A)\det(B)$.

C.  If $A$ is an element of $GL_R(3)$, then there exists a matrix $A^{-1}$ such that $A^{-1}A = I$.

D.  If $A$ is an element of $GL_R(3)$, then there exists a matrix $A$ such that $\det(A) \neq 1$.

It is given that the set of complex numbers $\mathbb{C}$ is a commutative ring with a multiplicative identity equal to 1. Let $z_1 = a + bi$ be any nonzero complex number. It can be shown that $\mathbb{C}$ is a field if there exists a $z_2 = x + yi$ satisfying which of the following equations?

A.  $\begin{bmatrix} a & b \\ b & a \end{bmatrix}\begin{bmatrix} x \\ y \end{bmatrix} = \begin{bmatrix} 1 \\ 0 \end{bmatrix}$

B.  $\begin{bmatrix} -a & b \\ -b & a \end{bmatrix}\begin{bmatrix} x \\ y \end{bmatrix} = \begin{bmatrix} 1 \\ 0 \end{bmatrix}$

C.  $\begin{bmatrix} -a & b \\ b & -a \end{bmatrix}\begin{bmatrix} x \\ y \end{bmatrix} = \begin{bmatrix} 1 \\ 0 \end{bmatrix}$

D.  $\begin{bmatrix} a & -b \\ b & a \end{bmatrix}\begin{bmatrix} x \\ y \end{bmatrix} = \begin{bmatrix} 1 \\ 0 \end{bmatrix}$

(a) LaTeX example 1        (b) LaTeX example 2

Figure A18: Examples of misspecified data to be replicated in LaTeX

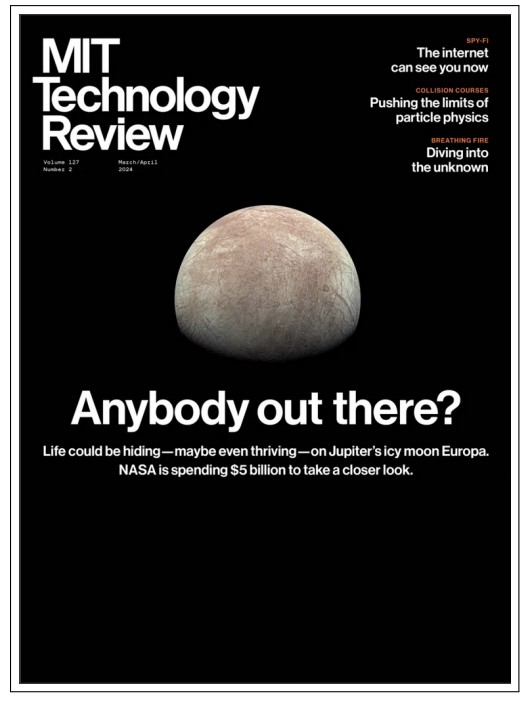
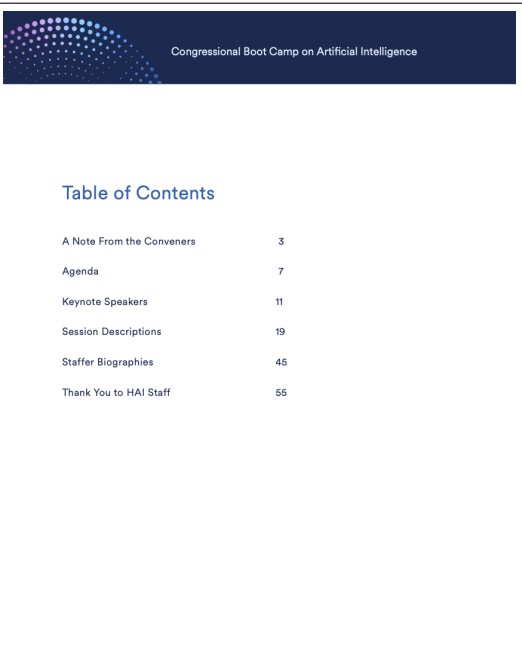

(a) Webpage example 1        (b) Webpage example 2

Figure A19: Examples of misspecified data to be replicated in HTML

# K    Detailed results on the misspecified data

Table A5: Average performances of VLMs on the 2 misspecified LaTeX instances. The scores reported are conditioned on successful rendering.

| Model | Compilation success | EMS | Pixel Similarity | CIS | LPIPS | SSIM | Edit Sim. |
|---|---|---|---|---|---|---|---|
| GPT-4o (2024-05-13) | **1** | **0.671** | 0.000 | **0.816** | 0.418 | 0.708 | – |
| IDEFICS-instruct (9B) | **1** | 0.644 | 0.000 | 0.771 | 0.421 | 0.739 | – |
| Claude 3 Opus (20240229) | **1** | 0.63 | 0.000 | 0.757 | 0.434 | 0.754 | – |
| Palmyra Vision 003 | **1** | 0.615 | 0.000 | 0.738 | **0.44** | **0.763** | – |
| Gemini 1.5 Pro (0409 preview) | **1** | 0.614 | 0.000 | 0.738 | **0.44** | **0.763** | – |
| Gemini 1.0 Pro Vision | 0.5 | 0.365 | 0.000 | 0.445 | 0.212 | 0.325 | – |
| Claude 3.5 Sonnet (20240620) | 0.5 | 0.353 | 0.000 | 0.402 | 0.174 | 0.364 | – |
| LLaVA 1.6 (13B) | 0.5 | 0.349 | 0.000 | 0.391 | 0.168 | 0.373 | – |
| Qwen-VL Chat | 0.5 | 0.343 | 0.000 | 0.385 | 0.107 | 0.376 | – |
| IDEFICS 2 (8B) | 0.5 | 0.343 | 0.000 | 0.385 | 0.170 | 0.376 | – |
| GPT-4V (1106 preview) | 0.5 | 0.332 | 0.000 | 0.386 | 0.186 | 0.388 | – |
| IDEFICS-instruct (80B) | 0.5 | 0.329 | 0.000 | 0.347 | 0.186 | 0.397 | – |
| Claude 3 Sonnet (20240229) | 0.5 | 0.324 | 0.000 | 0.346 | 0.187 | 0.398 | – |
| LLaVA 1.5 (13B) | 0 | 0 | 0.000 | 0.000 | 0.000 | 0.000 | – |

Table A6: Average performances of VLMs on the 2 webpage misspecified instances. The scores reported are conditioned on successful rendering.

| Model | Compilation success | EMS | Pixel Similarity | CIS | LPIPS | SSIM | Edit Sim. |
|---|---|---|---|---|---|---|---|
| Claude 3 Sonnet (20240229) | **1** | **0.735** | 0.000 | 0.757 | 0.326 | 0.704 | – |
| Claude 3.5 Sonnet (20240620) | **1** | 0.686 | 0.000 | 0.787 | 0.311 | 0.772 | – |
| Palmyra Vision 003 | **1** | 0.678 | 0.000 | 0.735 | 0.292 | 0.773 | – |
| GPT-4V (1106 preview) | **1** | 0.646 | 0.000 | **0.815** | 0.310 | 0.759 | – |
| GPT-4o (2024-05-13) | **1** | 0.642 | 0.000 | 0.768 | 0.312 | **0.778** | – |
| Gemini 1.5 Pro (0409 preview) | **1** | 0.611 | 0.000 | 0.783 | 0.314 | 0.771 | – |
| Claude 3 Opus (20240229) | **1** | 0.405 | 0.000 | 0.775 | **0.529** | 0.416 | – |
| LLaVA 1.5 (13B) | **1** | 0.163 | 0.000 | 0.705 | 0.414 | 0.46 | – |
| LLaVA 1.6 (13B) | 0.5 | 0.105 | 0.000 | 0.346 | 0.308 | 0.019 | – |
| Gemini 1.0 Pro Vision | 0.5 | 0.074 | 0.000 | 0.331 | 0.285 | 0.063 | – |
| Qwen-VL Chat | 0 | 0.000 | 0.000 | 0.000 | 0.000 | 0.000 | – |
| IDEFICS 2 (8B) | 0 | 0.000 | 0.000 | 0.000 | 0.000 | 0.000 | – |
| IDEFICS-instruct (9B) | 0 | 0.000 | 0.000 | 0.000 | 0.000 | 0.000 | – |
| IDEFICS-instruct (80B) | 0 | 0.000 | 0.000 | 0.000 | 0.000 | 0.000 | – |

# L Datasheet

## L.1 Motivation

Q1 **For what purpose was the dataset created?** *Was there a specific task in mind? Was there a specific gap that needed to be filled? Please provide a description.*

- The Image2Struct benchmark was created to evaluate the Vision-Language models on extracting structure from images. VLMs are prompted to generate the underlying structure (e.g., LaTeX code or HTML) from an input image (e.g., webpage screenshot). The structure is then rendered to produce an output image (e.g., rendered webpage), which is compared against the input image to produce a score.

Q2 **Who created the dataset (e.g., which team, research group) and on behalf of which entity (e.g., company, institution, organization)?**

- This benchmark is presented by the Center for Research on Foundation Models (CRFM), an interdisciplinary initiative born out of the Stanford Institute for Human-Centered Artificial Intelligence (HAI) that aims to make fundamental advances in the study, development, and deployment of foundation models. `https://crfm.stanford.edu/`.

Q3 **Who funded the creation of the dataset?** *If there is an associated grant, please provide the name of the grantor and the grant name and number.*

- None.

Q4 **Any other comments?**

- No.

## L.2 Composition

Q5 **What do the instances that comprise the dataset represent (e.g., documents, photos, people, countries)?** *Are there multiple types of instances (e.g., movies, users, and ratings; people and interactions between them; nodes and edges)? Please provide a description.*

- Image2Struct benchmark provides screenshots of webpages, sheet music, and scientific documents.

Q6 **How many instances are there in total (of each type, if appropriate)?**

- In all, we collected a total of 900 instances for webpages (300 each for HTML, CSS and JS), 1200 instances for LaTeX (300 each for equations, tables, algorithms, and plots), and 300 for music for a grand total of 2400 test instances.

Q7 **Does the dataset contain all possible instances or is it a sample (not necessarily random) of instances from a larger set?** *If the dataset is a sample, then what is the larger set? Is the sample representative of the larger set (e.g., geographic coverage)? If so, please describe how this representativeness was validated/verified. If it is not representative of the larger set, please describe why not (e.g., to cover a more diverse range of instances, because instances were withheld or unavailable).*

- It contains all the instances in this version of the dataset.

Q8 **What data does each instance consist of?** *"Raw" data (e.g., unprocessed text or images) or features? In either case, please provide a description.*

- screenshots of webpages, sheet music, and scientific documents.

Q9 **Is there a label or target associated with each instance?** *If so, please provide a description.*

- No, there is not label or target associated with each instance. Evaluation should be made using round-trip evaluation.

Q10 **Is any information missing from individual instances?** *If so, please provide a description, explaining why this information is missing (e.g., because it was unavailable). This does not include intentionally removed information, but might include, e.g., redacted text.*

- No except for wild instances that intentionally do not have any sources or metadata.

Q11 **Are relationships between individual instances made explicit (e.g., users' movie ratings, social network links)?** *If so, please describe how these relationships are made explicit.*

- Every instances has a link to its source.

Q12 **Are there recommended data splits (e.g., training, development/validation, testing)?** *If so, please provide a description of these splits, explaining the rationale behind them.*

- No.

Q13 **Are there any errors, sources of noise, or redundancies in the dataset?** *If so, please provide a description.*

- No.

Q14 **Is the dataset self-contained, or does it link to or otherwise rely on external resources (e.g., websites, tweets, other datasets)?** *If it links to or relies on external resources, a) are there guarantees that they will exist, and remain constant, over time; b) are there official archival versions of the complete dataset (i.e., including the external resources as they existed at the time the dataset was created); c) are there any restrictions (e.g., licenses, fees) associated with any of the external resources that might apply to a future user? Please provide descriptions of all external resources and any restrictions associated with them, as well as links or other access points, as appropriate.*

- The dataset is self-contained. Everything is available at `https://huggingface.co/datasets/stanford-crfm/image2struct-webpage-v1` (Webpages), `https://huggingface.co/datasets/stanford-crfm/image2struct-latex-v1` (LaTeX), and `https://huggingface.co/datasets/stanford-crfm/image2struct-musicsheet-v1` (Musical Scores).

Q15 **Does the dataset contain data that might be considered confidential (e.g., data that is protected by legal privilege or by doctor–patient confidentiality, data that includes the content of individuals' non-public communications)?** *If so, please provide a description.*

- No. All the instances are obtained from Github, arXiv, or IMSLP.

Q16 **Does the dataset contain data that, if viewed directly, might be offensive, insulting, threatening, or might otherwise cause anxiety?** *If so, please describe why.*

- We apply automated filters via the Perspective API to remove inappropriate content to the best of our abilities.

Q17 **Does the dataset relate to people?** *If not, you may skip the remaining questions in this section.*

- No. However, people may be present in screenshots if some websites contain mages of people in them.

Q18 **Does the dataset identify any subpopulations (e.g., by age, gender)?**

- No.

Q19 **Is it possible to identify individuals (i.e., one or more natural persons), either directly or indirectly (i.e., in combination with other data) from the dataset?** *If so, please describe how.*

- It may be possible to identify individuals if the data instances is linked to the profile on online communities.

Q20 **Does the dataset contain data that might be considered sensitive in any way (e.g., data that reveals racial or ethnic origins, sexual orientations, religious beliefs, political opinions or union memberships, or locations; financial or health data; biometric or genetic data; forms of government identification, such as social security numbers; criminal history)?** *If so, please provide a description.*

- No. We have removed inappropriate content to the best of our abilities.

Q21 **Any other comments?**

- We caution discretion on behalf of the user and call for responsible usage of the benchmark for research purposes only.

### L.3 Collection Process

**Q22 How was the data associated with each instance acquired?** *Was the data directly observable (e.g., raw text, movie ratings), reported by subjects (e.g., survey responses), or indirectly inferred/derived from other data (e.g., part-of-speech tags, model-based guesses for age or language)? If data was reported by subjects or indirectly inferred/derived from other data, was the data validated/verified? If so, please describe how.*

- All the data in this dataset was directly observable and collected from public sources namely Github, arXiv, and IMSLP. The data was collected based on some search criterias such as categories and dates. The data was then fetched according to the search results order of these websites and manually inspected to control the quality and diversity of the dataset.

**Q23 What mechanisms or procedures were used to collect the data (e.g., hardware apparatus or sensor, manual human curation, software program, software API)?** *How were these mechanisms or procedures validated?*

- The existing scenarios were downloaded by us.
- The data was collected using software API or by scrapping the aforementioned websites. Manual inspection used to change the search criterias but no manuel filtering was performed.

**Q24 If the dataset is a sample from a larger set, what was the sampling strategy (e.g., deterministic, probabilistic with specific sampling probabilities)?**

- We use the whole datasets

**Q25 Who was involved in the data collection process (e.g., students, crowdworkers, contractors) and how were they compensated (e.g., how much were crowdworkers paid)?**

- The authors of this paper collected the scenarios.
- Data was collected by a script.

**Q26 Over what timeframe was the data collected? Does this timeframe match the creation timeframe of the data associated with the instances (e.g., recent crawl of old news articles)?** *If not, please describe the timeframe in which the data associated with the instances was created.*

- The data was collected in March 2024 and the data is from January and February 2024.

**Q27 Were any ethical review processes conducted (e.g., by an institutional review board)?** *If so, please provide a description of these review processes, including the outcomes, as well as a link or other access point to any supporting documentation.*

- No.

**Q28 Does the dataset relate to people?** *If not, you may skip the remaining questions in this section.*

- People may very rarely appear in some images of some webpages, although they are not the focus of the dataset.

**Q29 Did you collect the data from the individuals in question directly, or obtain it via third parties or other sources (e.g., websites)?**

- NA.

**Q30 Were the individuals in question notified about the data collection?** *If so, please describe (or show with screenshots or other information) how notice was provided, and provide a link or other access point to, or otherwise reproduce, the exact language of the notification itself.*

- NA.

**Q31 Did the individuals in question consent to the collection and use of their data?** *If so, please describe (or show with screenshots or other information) how consent was requested and provided, and provide a link or other access point to, or otherwise reproduce, the exact language to which the individuals consented.*

- NA.

Q32 **If consent was obtained, were the consenting individuals provided with a mechanism to revoke their consent in the future or for certain uses?** *If so, please provide a description, as well as a link or other access point to the mechanism (if appropriate).*

- NA.

Q33 **Has an analysis of the potential impact of the dataset and its use on data subjects (e.g., a data protection impact analysis) been conducted?** *If so, please provide a description of this analysis, including the outcomes, as well as a link or other access point to any supporting documentation.*

- NA.

Q34 **Any other comments?**

- NA.

## L.4 Preprocessing, Cleaning, and/or Labeling

Q35 **Was any preprocessing/cleaning/labeling of the data done (e.g., discretization or bucketing, tokenization, part-of-speech tagging, SIFT feature extraction, removal of instances, processing of missing values)?** *If so, please provide a description. If not, you may skip the remainder of the questions in this section.*

- Yes. For LaTeX we extract from the original LaTeX code some self-sufficient blocks (equations, plots, ...) and add our own wrappers around it *(Please refer to our Github repository to see the packages we use as this may change in the future)*. For musical scores we extract measures from pages, please refer to Section 2.1.3 for more details.

Q36 **Was the "raw" data saved in addition to the preprocessed/cleaned/labeled data (e.g., to support unanticipated future uses)?** *If so, please provide a link or other access point to the "raw" data.*

- No but the link to the originial data is provided.

Q37 **Is the software used to preprocess/clean/label the instances available?** *If so, please provide a link or other access point.*

- Yes, it is part of our Github repository: `https://github.com/stanford-crfm/image2struct/`.

Q38 **Any other comments?**

- No.

## L.5 Uses

Q39 **Has the dataset been used for any tasks already?** *If so, please provide a description.*

- Not yet. Image2Struct is a new benchmark.

Q40 **Is there a repository that links to any or all papers or systems that use the dataset?** *If so, please provide a link or other access point.*

- We will provide links to works that use our benchmark at `https://crfm.stanford.edu/helm/image2struct/latest/`.

Q41 **What (other) tasks could the dataset be used for?**

- The primary use case of our benchmark is the evluation of VLMs in the context of extracting information from images.

Q42 **Is there anything about the composition of the dataset or the way it was collected and preprocessed/cleaned/labeled that might impact future uses?** *For example, is there anything that a future user might need to know to avoid uses that could result in unfair treatment of individuals or groups (e.g., stereotyping, quality of service issues) or other undesirable harms (e.g., financial harms, legal risks) If so, please provide a description. Is there anything a future user could do to mitigate these undesirable harms?*

- Our benchmark contains contents uploaded by individual users. While we try our best to filter any inappropriate content, we cannot guarantee that our dataset is perfectly non-biased and unharmful.

Q43 **Are there tasks for which the dataset should not be used?** *If so, please provide a description.*

- This dataset should not be used to train new models in order to keep this evaluation fair.
- This benchmark should not be used to aid in military or surveillance tasks.

Q44 **Any other comments?**

- No.

## L.6 Distribution and License

Q45 **Will the dataset be distributed to third parties outside of the entity (e.g., company, institution, organization) on behalf of which the dataset was created?** *If so, please provide a description.*

- Yes, this benchmark will be open-source.

Q46 **How will the dataset be distributed (e.g., tarball on website, API, GitHub)?** *Does the dataset have a digital object identifier (DOI)?*

- Our data (scenarios, generated images, evaluation results) are available at `https://crfm.stanford.edu/helm/image2struct/latest/`.
- Our code used for evaluation is available at `https://github.com/stanford-crfm/image2struct`.

Q47 **When will the dataset be distributed?**

- April 1, 2024 and onward.

Q48 **Will the dataset be distributed under a copyright or other intellectual property (IP) license, and/or under applicable terms of use (ToU)?** *If so, please describe this license and/or ToU, and provide a link or other access point to, or otherwise reproduce, any relevant licensing terms or ToU, as well as any fees associated with these restrictions.*

- Our code is released under the **Apache-2.0** license

Q49 **Have any third parties imposed IP-based or other restrictions on the data associated with the instances?** *If so, please describe these restrictions, and provide a link or other access point to, or otherwise reproduce, any relevant licensing terms, as well as any fees associated with these restrictions.*

- We own the metadata and release as CC-BY-4.0.
- We do not own the copyright of the images or text.

Q50 **Do any export controls or other regulatory restrictions apply to the dataset or to individual instances?** *If so, please describe these restrictions, and provide a link or other access point to, or otherwise reproduce, any supporting documentation.*

- No.

Q51 **Any other comments?**

- No.

## L.7 Maintenance

Q52 **Who will be supporting/hosting/maintaining the dataset?**

- Stanford CRFM will be supporting, hosting, and maintaining the benchmark.

Q53 **How can the owner/curator/manager of the dataset be contacted (e.g., email address)?**

- `https://crfm.stanford.edu`

Q54 **Is there an erratum?** *If so, please provide a link or other access point.*

- There is no erratum for our initial release. Errata will be documented as future releases on the benchmark website.

Q55 **Will the dataset be updated (e.g., to correct labeling errors, add new instances, delete instances)?** *If so, please describe how often, by whom, and how updates will be communicated to users (e.g., mailing list, GitHub)?*

- Image2Struct will be updated. We plan to expand scenarios, metrics, and models to be evaluated.

Q56 **If the dataset relates to people, are there applicable limits on the retention of the data associated with the instances (e.g., were individuals in question told that their data would be retained for a fixed period of time and then deleted)?** *If so, please describe these limits and explain how they will be enforced.*

- People may contact us at `https://crfm.stanford.edu` to add specific samples to a blacklist.

Q57 **Will older versions of the dataset continue to be supported/hosted/maintained?** *If so, please describe how. If not, please describe how its obsolescence will be communicated to users.*

- We will host other versions. We plan to rerun the data collection on a regular basis to ensure that some unseen data is always available.

Q58 **If others want to extend/augment/build on/contribute to the dataset, is there a mechanism for them to do so?** *If so, please provide a description. Will these contributions be validated/verified? If so, please describe how. If not, why not? Is there a process for communicating/distributing these contributions to other users? If so, please provide a description.*

- People may contact us at `https://crfm.stanford.edu` to request adding new instances, metrics, or models.

Q59 **Any other comments?**

- No.

