# Image2Struct - Supplementary materials

## Links

**Datasets:**
- Image2Latex: https://huggingface.co/datasets/stanford-crfm/i2s-latex
- Image2Webpage: https://huggingface.co/datasets/stanford-crfm/i2s-webpage
- Image2MusicSheet: https://huggingface.co/datasets/stanford-crfm/i2s-musicsheet

**Benchmark results/leaderboard:** https://crfm.stanford.edu/helm/image2structure/latest/

**Code:**
- Data collection: https://github.com/stanford-crfm/image2structure
- Evaluation: https://github.com/stanford-crfm/helm

**Maintenance:** Josselin Somerville Roberts: https://josselinsomervilleroberts.github.io/

**License:** Apache License
  Version 2.0, January 2004
  http://www.apache.org/licenses/

## Croissant URL to get Dataset metadata

- Image2Latex:
  https://huggingface.co/api/datasets/stanford-crfm/i2s-musicsheet/croissant
- Image2Webpage: https://huggingface.co/api/datasets/stanford-crfm/i2s-webpage/croissant
- Image2Musicsheet: https://huggingface.co/api/datasets/stanford-crfm/i2s-musicsheet/croissant

# Datasheet for datasets

This section details the contents of one row in the dataset. Some fields are present across all data types and all subsets and are denoted as "required". Other fields depend on the data type and might be present or not depending on if there is a ground truth or not.

This structure has been chosen to be easily extendable. There is for example the presence of "assets" that are not used in the paper but could be used to save images that are required to render figure in Latex for example.

Dataset contents:
- image (image - required): encoding of an image
- structure (Any - optional): original structure that generated the image. Can be anything *(In our case it is text for Latex and base64 encoding of a zipped folder for webpages)*
- text (String - optional): text extracted from the structure
- download_url (String - required): URL where the data comes from
- instance_name (String - required): represents the instance (internal use)
- date (String - required): Date of the data (usually the date when it was uploaded to the site we scrapped from so the data could technically be older if it is re-uploaded)
- date_scrapped (String - required): Date when the data was collected
- additional_info (Dict[str, Any] - required): Contains some metadata that depends on the data type. Can contain: title, page count, id, …
- compilation_info (Dict[str, Any] - required): Metadata of the compilation
- rendering_filters (Dict[str, Any] - optional): Metadata of the filtering applied to the rendered image
- file_filters (Dict[str, Any] - optional): Metadata of the filtering applied to the downloaded data from the scrapped website
- assets (List[Any] - optional): List of base 64 encoded assets.
- uuid (String - required): Unique identifier
- length (int - optional): Some measure of data quantity (depends on data type)
- difficulty (String - required): Some measure of difficulty (depends on data type)

| Dataset | Image2 Latex | Image2 Webpage | Image2 Musicsheet | Image2 Latex - *Wild* | Image2 Webpage - *Wild* |
|---|---|---|---|---|---|
| Subsets | equation, table, algorithm, plot | CSS, HTML, Javascript | music | wild *(equation\*)* | wild |
| Difficulties | Easy, Medium, Hard | Easy, Medium, Hard | Easy, Medium, Hard | Hard | Hard |
| Number of instances *(per subset)* | 300 | 300 | 300 | 50 | 50 |
| All instances? | Yes | Yes | Yes | Yes | Yes |
| Automatically renewable? | Yes | Yes | Yes | No | No |
| Ground Truth ("structure") | Yes | Yes | No | No | No |
| Extracted text | Yes | Yes | No | No | No |
| Data splits | 1 split only: validation | 1 split only: validation | 1 split only: validation | 1 split only: validation | 1 split only: validation |
| Data source | Arxiv | GitHub | IMLSP | Wikipedia | *See Annex* |

*\* The wild set of Image2Latex only contains equations*

## Author Statement

I, Josselin Somerville Roberts, on behalf of all the authors of this paper, hereby declare that I bear all responsibility for any violations of rights that may arise from the contents of this supplementary material. Furthermore, I confirm that all data included herein is properly licensed and any necessary permissions for use and publication have been obtained.

Josselin Somerville Roberts on behalf of all the authors of this paper.