# OpenReview forum: "Image2Struct: Benchmarking Structure Extraction for Vision-Language Models"
_NeurIPS.cc/2024/Datasets_and_Benchmarks_Track — NeurIPS 2024 Track Datasets and Benchmarks Poster_

### Official Review · Reviewer_khfM · 2024-07-23
**Innovative evaluation protocol and new benchmark**

**Rating:** 7
**Confidence:** 3

**Review:**

This paper does one thing and it does it really well: An automatic benchmark and evaluation protocol that fulfills several non-trivial requirements such as drawing fresh data and live-screenshotting. It doesn’t try to do more but instead really goes in-depth motivating and analyzing Image2Struct.
It could introduce or discuss the metrics a bit better and quantify some aspects of them.

**Strengths:**

The paper demonstrates deep knowledge and intuition for the current evaluation landscape of (V)LLMs, beyond static benchmarks: data leakage, distribution shifts over time, leaderboard maintenance.
It also strikes a nice balance between testing for complex long outputs and automatic objective evaluation.

On the engineering side of things, the whole pipeline is impressive, i.e. on the fly compilation and screenshotting when someone submits to the leaderboard.

The analyses in the discussion section make sense and shed some light on questions that came up for me when reading previous sections.

**Additional Feedback:**

Line 76: close bracket.

In the tables it is helpful to add upwards/downwards arrows to indicate if performance is better when the number is higher/lower.

How do you compare models if it is always run on new data?

**Clarity:**

Overall the paper is very easy to follow and the first sentence of the abstract sets the tone of what to expect and it meets those expectations.

On a more detailed level, the EMD metric is hard to follow and should be motivated and illustrated on a high level first.

**Correctness:**

Is the FID score in the paper really an FID score and not just akin to a cosine similarity? FID scores usually consider distributions of images, no? Please clarify or change this section.

**Documentation:**

The pipeline is well documented with sufficient detail.

The Datasheet does not include all the questions that are usually in it, as far as I can tell?

“The data is filtered for relevance, diversity, and toxicity after being downloaded to ensure quality
The Perspective API is deployed to filter toxic content” → this is described in good detail but could be expanded on more in the appendix.

**Ethics:**

No issues.

**Limitations:**

Limitations and broader impacts are discussed clearly.

**Opportunities For Improvement:**

To bridge the gap a little towards the classic sense of “vision and language” task, e.g. “a ball next to a tree”, is it possible to have the same setup on simulation engines → scene graph → rendering or something similar? This would make the benchmark a bit more diverse and cover classic shortcomings issues identified in VL such as spatial reasoning and visual compositionality.This is not a major criticism, but mostly a suggestion.

Arguably, the scientifically most challenging part is the “Image similarity metrics”, which is done reasonably well but also needs a bit more clarity and intuition (see below). Human correlation studies could’ve helped to justify them. The authors use correlation to edit-distance which makes sense. One question I have then: Why even compile to images and screenshot if staying in the text space is best? The way you scrape, do you not always have access to the source code? I see that you discuss this and argue that this is more flexible when no source code is available, so this is only a minor criticism/question!

Alternatively, hand-annotating e.g. 100 model outputs to see how well models do based on an easy to follow metric (=human judgment) would be helpful. Right now it is unclear how well models are doing: win rates are relative and it is not clear what an automatic metric number really means. All we can really know is the ordering of models.
Such hand annotated examples could justify claims such as line 251 “ Music -where VLMs produce few, if any, good results” compared to the automatic metric where numbers could differ for reasons other than task difficulty.

**Relation To Prior Work:**

The related work section is very contextual (in a good way) and directly ties to the main story of the paper! It is nice when it doesn’t break the flow of the paper and does not just list similar papers generically.

**Summary And Contributions:**

Image2Struct is a benchmark and evaluation protocol for models to translate an image (e.g. screenshot of Latex) into the underlying structure (Latex code). The authors propose a round-trip/cyclic evaluation setup that enables automatic evaluation and cases where several solutions are equally correct.
The authors also innovate by continuously drawing fresh data every time the benchmark is run. For example, this reduces data leakage issues from training future models.
The benchmark tests specifically for three categories of skills: mathematical formulas, web page code and musical notes. When someone submits to the leaderboard it automatically finds new examples from websites (i.e. arxiv) and does all the compilation and screenshotting on the fly.
The authors then test recent VLMs (ChatGPT-V, Gemini, … and variations of them, for example) and find that models still have several failure modes.

---

> ### Author Rebuttal · Authors · 2024-08-17
>
> We thank you for both taking the time to review our work and for your kind comments. We address your questions point-by-point as follows:
>
> > To bridge the gap a little towards the classic sense of “vision and language” task, e.g. “a ball next to a tree”, is it possible to have the same setup on simulation engines → scene graph → rendering or something similar? This would make the benchmark a bit more diverse and cover classic shortcomings issues identified in VL such as spatial reasoning and visual compositionality. This is not a major criticism, but mostly a suggestion.
>
> We noted in our `Broader Impact’ section that “beyond parsing images, we envision using the same framework to extract information from multimodal documents (e.g., electronic health records), help robots to replicate a physical scene given an image, or to replicate the trading strategy of a systematic fund given the time series of its profit and loss’’ (Lines 274–277). One idea that we had was to feed VLMs an image of a 3D scene, have them generate the codes (e.g., in UnReal Engine), and then take a screenshot of the rendered scene. We will explore this further in future works.
>
> > Arguably, the scientifically most challenging part is the “Image similarity metrics”, which is done reasonably well but also needs a bit more clarity and intuition (see below). Human correlation studies could’ve helped to justify them. The authors use correlation to edit-distance which makes sense. One question I have then: Why even compile to images and screenshot if staying in the text space is best? The way you scrape, do you not always have access to the source code? I see that you discuss this and argue that this is more flexible when no source code is available, so this is only a minor criticism/question!
>
> Thank you for this question. We started off with the idea that VLMs should be able to extract structure from images which can be recompiled and compared against the source image. We tested some image similarity metrics but were not sure which was better. That was when we had the idea of correlating them with the edit distance between the ground-truth source code and the predicted structure. We had no expectation before we ran the experiments and the good results were simply a validation of the metrics; with this, we are more confident that the round-trip comparison can be used for instances where we do not have ground-truth structures. We hope this answers your question.
>
> > Alternatively, hand-annotating e.g. 100 model outputs to see how well models do based on an easy to follow metric (=human judgment) would be helpful. Right now it is unclear how well models are doing: win rates are relative and it is not clear what an automatic metric number really means. All we can really know is the ordering of models. Such hand annotated examples could justify claims such as line 251 “ Music -where VLMs produce few, if any, good results” compared to the automatic metric where numbers could differ for reasons other than task difficulty.
>
> We are currently running an experiment where we manually label randomly selected examples against predefined rubrics to analyze the errors made and compare the human annotations against the image metrics. We will report the results in the camera-ready version.
>
> > Is the FID score in the paper really an FID score and not just akin to a cosine similarity? FID scores usually consider distributions of images, no? Please clarify or change this section.
>
> Thank you for the feedback. It is indeed the cosine distance and we have changed it to ‘Cosine similarity between Inception vectors (CIS)’ in our latest manuscript.
>
> > On a more detailed level, the EMD metric is hard to follow and should be motivated and illustrated on a high level first.
>
> We have added a detailed explanation of block-EMD and EMS in our draft. The extracted section is attached in this rebuttal (**see attached PDF**). We hope that the rewritten section will clear up any misunderstanding.
>
> > The Datasheet does not include all the questions that are usually in it, as far as I can tell?
>
> We have updated the datasheet and added the 59 questions suggested in Datasheets for Datasets: https://arxiv.org/abs/1803.09010 (not included here because of maximum character constraints). Additionally, we have added the full Datasheet with the questions, links, metadata and all the other information in "Appendix G. Datasheet".
>
> > “The data is filtered for relevance, diversity, and toxicity after being downloaded to ensure quality The Perspective API is deployed to filter toxic content” → this is described in good detail but could be expanded on more in the appendix.
>
> We have added more detailed information about the data collection and filtering in sections 2.1.1-2.1.3. We state a few sentences from our edited manuscript that address this concern:
>
> - For all the domains, “we collect 300 valid test instances per subdomain with a maximum of 40 instances on a single day in order to induce temporal diversity.”
> - For Image2Latex, “data points where the rendered image is mostly blank are discarded”.
> - Also for Image2Latex, “we select the final data set in a way that balances the number of instances per subject (e.g., physics, maths, or economics) and structures (e.g., equation or algorithm)”.
> - For Image2Webpage, we state that “screenshots that are completely or nearly completely white are removed from the dataset and the remaining images are de-duplicated.”
>
> > In the tables it is helpful to add upwards/downwards arrows to indicate if performance is better when the number is higher/lower.
>
> Thank you. We have added the arrows in our manuscript.
>
> > How do you compare models if it is always run on new data?
>
> One possibility would be that all the models are tested on the new data and the scores for the models across time can be smoothed using an exponentially-weighted average.

---

> > ### Comment · Reviewer_khfM · 2024-08-19
> >
> > Thank you for addressing my questions! Most of it is now resolved from my side. There is only a smaller comment/clarification regarding EMS:
> >
> > I am still a bit unclear on why EMS is needed: It seems the cosine distance correlates equally well with edit distance but is conceptually much simpler. So, on a high level it is not clear why it is intuitively needed, compared to the simpler ones.

---

> > > ### Author Rebuttal · Authors · 2024-08-21
> > >
> > > Thank you once again for helping to strengthen our work.
> > >
> > > We will emphasize that the preferred metric depends on the use-case. For example, the cosine similarity uses a neural network and is hence less interpretable compared to EMS, where matched patches can be visualized and inspected. We give results for multiple commonly-used metrics so that users can decide which metric they want to use when interpreting the results from the benchmark.

---

### Official Review · Reviewer_qbWL · 2024-07-26
**A novel form of evaluation benchmark for VLM**

**Rating:** 7
**Confidence:** 4

**Review:**

Pros:
- Transparency. The dataset, website and document are currently available and transparent. The information disclosed already shows the important information regarding this benchmark that we may care about. From the materials currently released, it can be used and followed by other researchers in the community.
- Novelty. The idea of generating images with the extracted code and make comparison with the original images, is quite novel, elegant, and smart. It is different to most of the existing benchmarks with human annotations.
- Potential of the dataset to keep being updated. As the benchmark does not require human annotation, it becomes easier to keeping it updated, which makes it a fresh and solid benchmark for comparing methods that is difficult to game.
- Potential impact to MLLM community.

Cons:
- The correctness of the proposed evaluation strategy. I have some concern over this.
  - If the extracted code contains a few small, forgivable grammatical error that may result in compiling error, does the model fail completely on this case? That seems not reasonable.
  - For images containing characters and symbols, some textual differences may results in slight visual differences. Thus, the visual similarity may not be ideal to reflect the actual error of the extracted structural information.

**Strengths:**

See the pros above.

**Additional Feedback:**

I like this work and I think it will have some impact to the community. Happy to be a reviewer for this work. I hope my concerns in the cons above may be answered in the rebuttal. Thanks in advance.

**Clarity:**

Generally, yes. The paper is well written and clear. The demonstration is smooth. The figures are simple and easy to understand.

**Correctness:**

Generally yes. I have some concern over the image similarity-based evaluation metric for reflecting the textual differences. Please see the cons above.

**Documentation:**

Yes. The documents are provided. The URL is available. A new version (1.0.1) is already provided, which is an update of the submitted 1.0.0 version. The availability, maintenance and reproducibility of this benchmark is trustworthy, in my personal opinion.

**Ethics:**

I do not find any ethical concerns needed to be raised, considering the benchmark contains images from open websites like arxiv.

**Limitations:**

See the cons above.

**Opportunities For Improvement:**

The image similarity based metric may not be accurate to demonstrate the textual differences. I hope more clarification on this can be provided. See the cons above.

**Relation To Prior Work:**

Yes.

**Summary And Contributions:**

- This paper propose a new evaluation benchmark for MLLMs extracting structural information from images.
- Compared with existing benchmarks, the proposed benchmark does not compare the output with human annotated ground truth. Instead, it compares the image generated via the code extracted by MLLM with the original image.
- The authors evaluate multiple important and recent MLLMs, and provides a leaderboard, which they will continue to update.

---

> ### Author Rebuttal · Authors · 2024-08-17
>
> Thank you for taking the time to review the paper and for your suggestions that strengthen our paper.
>
> > If the extracted code contains a few small, forgivable grammatical error that may result in compiling error, does the model fail completely on this case? That seems not reasonable.
>
> We have a post-processing pipeline to maximize the chances of compilation. We have added our post-processing steps to our latest draft, which we replicate in the attached PDF file.
>
> > For images containing characters and symbols, some textual differences may results in slight visual differences. Thus, the visual similarity may not be ideal to reflect the actual error of the extracted structural information.
>
> Minor changes in the characters and symbols should not significantly change the visual display. We give an example where some models “produce an overline in place of an overbar” (Lines 224–225). To further quantify this, we are currently running an experiment where we manually label randomly selected examples against predefined rubrics to analyze the errors made and compare the human annotations against the image metrics. We will report the results in the camera-ready version.
>
> Once again, thank you for your feedback and we will be happy to address any further concerns.

---

### Official Review · Reviewer_ksMf · 2024-07-30
**VLMs struggle to extract structured information from images. A good benchmark?**

**Rating:** 6
**Confidence:** 4
**Clarity:** Yes.

**Review:**

The paper proposes a novel and original benchmark for the evaluation of generative VLMs on the task of structured information extraction from images. Structure is defined in terms of code which can be compiled by a task-specific compiler to render the original image. It focuses on the tasks of code generation from screenshots of webpages, music sheets, and latex documents (algorithms, equations, plots, or tables). The paper addresses an important problem of relevance to the community, is written well, and shows via experiments that both closed-API and open source VLMs struggle to perform well on these tasks, the former significantly outsourcing the latter ones. The benchmark has the potential for significant impact on spurring research on this important topic.

However, I have concerns regarding the metrics used for automatic, closed-loop evaluation of VLMs and hence fear that using these metrics for ranking the models may not be proper. In addition, the paper needs to have a more comprehensive survey of the literature.

The above strengths (pros) and limitations (cons) are discussed more in detail below.

===
**Post-rebuttal comment**: Some of my concerns regarding the paper have been alleviated and I'm increased by score accordingly.

**Strengths:**

- **Relevance**: The paper addresses a very important problem in an area of increasing concern and relevance (Right to be Forgotten). The paper should be of interest that transcends the vision & language community to the broader research community.
- **Clarity**: Paper is written well.
- **Originality/ Novelty**: This is a first benchmark on the above tasks which is also designed for automatic closed-loop evaluation of new, unseen data from the real-world that may be continuously/ periodically scraped so that it can’t be ‘gamed’ by VLMs.
  - Struct2Image is a novel benchmark that creates a dataset for 8 Tasks across 3 subdomains: sheet music, web pages (CSS, HTML, JS), and LaTex document components (Algorithm, Equation, Plot, Table).
  - Performance of 12 popular VLMs is benchmarked. These include those in the Claude, Sonnet, GPT-4, Gemini, LLaVa, IDEFICS and Qwen-WL family on the above data sets.
- **Reproducibility**: The URL to the project page with a leaderboard, Github code repository and the datasets is shared. This will aid in reproducing the results.
- **Significance**: Image2Struct demonstrates that VLMs don’t perform well on the above tasks. Significant improvement is needed before these models are able to solve these tasks. The paper has a good potential for spurring research on this important topic. Its main takeaways are,
  - None of the models can interpret sheet music.
  - They are able to extract the elements but unable to puck on visual nuances.
  - Closed-API versions significantly outperform open-source ones.
  - Different VLMs perform better at different tasks.
  - GPT-4 Omni has the highest win rate and most consistent across the board performance.

**Additional Feedback:**

Beyond the main limitations shared in ‘Opportunities for Improvement’, I have the following comments/ queries addressing which will improve the quality of the paper.

1. Multiple queries (line. 203-206) may improve the compilation success rates and hence may significantly impact the results.
2. The source image may have secondary information (background, fonts and font size, and other stylistic information) which the VLMs may find hard to extract which may make it difficult to use predefined metrics in a meaningful way.
3. Examples of “multiple correct answers” (lines 7-9) should be shared.
4. It will be good to evaluate the metrics themselves on a separately validated, perhaps manually curated, dataset containing ‘correct’ extractions.
5. Metric distributions for both good code extraction and erroneous ones should be created and compared. This will allow a better utilization (and appropriate improvement) of the metrics for scoring and ranking models.
6. Was a quality check done after the preprocessing steps?
7. The definition of Block-EMD is not clear. Is it on a block of the screenshot or something else (‘patches of pixels’ on line 181)?
8. The normalization in  (2) needs to be clarified.
9. Is the final EMS score summed over all the blocks?
10. What is the size of the dataset?

**Correctness:**

I have misgivings regarding the metrics used and the ranking and takeaways based on them (see limitations).

**Documentation:**

Yes.

**Ethics:**

No.

**Limitations:**

Yes.

**Opportunities For Improvement:**

**Literature Survey**:  Important related art is missing. While some work below may be concurrent, it’s useful to contextualize the presented work.
- **General Code Generation literature (LLMs)**:
  - DeepSeek-Coder (Guo et al, arXiv preprint arXiv:2401.14196, https://github.com/deepseek-ai/DeepSeek-Coder)
  - StarCoder (Li et al, TMLR’23, https://github.com/bigcode-project/starcoder)
  - Code LLaMA (Roziere et al, arXiv preprint arXiv:2308.12950, 2023 ),
- **Multimodal Code generation**
  - MMCode (Li et al, arXiv preprint arXiv:2404.09486, 2024),https://github.com/happylkx/MMCode, https://huggingface.co/datasets/likaixin/MMCode )
  - Design2Code (Si et al, arXiv preprint arXiv:2403.03163, 2024, https://salt-nlp.github.io/Design2Code/ https://github.com/NoviScl/Design2Code https://huggingface.co/datasets/SALT-NLP/Design2Code )
  - UI-to-Code (Soselia et al, https://arxiv.org/abs/2305.14637 )
  - Plot2Code Benchmark (https://huggingface.co/datasets/TencentARC/Plot2Code https://github.com/TencentARC/Plot2Code https://arxiv.org/abs/2405.07990 )
  - **UI Screenshots to Code**: Pix2Code (Beltramelli, 2018), Wan et al (arXiv:2406.16386v1) etc.
- **Image Similarity Metrics**: There is a rich literature on image similarity metrics and this aspect is really weak in the paper. Examples include SSIM, LPIPS, and those based on siamese networks and triplet networks for face recognition (see DeepFace) as well as for image retrieval.

**Technical approach**: The capability of automatic closed-loop evaluation is dependent on sound metrics. Image metrics are sought to surrogate for the Levenshtein (edit distance) metric when the GT code is not available, which is expected to be the case for a live, continuously changing benchmark.
- L1, pw-FID, and even EMS don’t evoke confidence as good metrics that can surrogate well for the Levenshtein metric or are good from a first-principle perspective, to benchmark Image2Struct models. In fact, Figure A3 and A6 suggest that EMS isn’t a good metric.
- In such a scenario, it is hard to trust EMS to provide a good performance ranking mechanism either. This puts a question mark on the results shared in the paper.

**Relation To Prior Work:**

Yes but important art is missing (see above).

**Summary And Contributions:**

The paper introduces the Image2Struct benchmark to evaluate VLMs on the task of extracting structured information from images. The benchmark is designed for automatic closed-loop evaluation of new, unseen data from the real-world that may be continuously/ periodically scraped so that it can’t be ‘gamed’ by VLMs. The benchmark is constructed on three tasks – generation of compilable code from screenshots of webpages (to HTML, CSS, JS code), music sheets (to Lilypond code), or latex algorithms, equations, plots or tables (to latex code). Six SOTA closed-API and six open source VLM models are evaluated. The benchmark shows that VLMs don’t perform well on these tasks in general and that there is significant room for improvement before these models are able to solve these tasks.

---

> ### Author Rebuttal · Authors · 2024-08-17
>
> Thank you for taking the time to review our paper and for providing valuable feedback. We address your concerns point-by-point as follows:
>
> >  Important related art is missing. While some work below may be concurrent, it’s useful to contextualize the presented work.
>
> Thank you for your feedback. We have incorporated your suggestions and added these literature in the camera-ready version. We also implemented several of the image similarity metrics such as SSIM and LPIPS. These numbers have been added to Appendix C (detailed results) and are already available in the new run online: https://crfm.stanford.edu/helm/image2struct/v1.0.1/#/leaderboard
>
> > The capability of automatic closed-loop evaluation is dependent on sound metrics. Image metrics are sought to surrogate for the Levenshtein (edit distance) metric when the GT code is not available, which is expected to be the case for a live, continuously changing benchmark.
>
> Thank you for your feedback. We noted in our submitted manuscript that “the metrics introduced in this paper are not perfect even though they have high correlation with the edit distance between the generated and ground-truth source code. Our work opens up future research directions for the development of methods that can evaluate the degree of similarity between rendered and ground-truth images” [Lines 268–271]. In addition, for the camera-ready version, we have added LPIPS and SSIM as metrics. This can be viewed online at https://crfm.stanford.edu/helm/image2struct/latest/#/
>
> In order to build confidence, we will randomly select VLM output, rate the VLM output manually, and report the results in the camera-ready version. In addition, taking your suggestions, we will create and compare the metric distributions for a small samples of both good code extraction and erroneous ones.
>
> > Multiple queries (line. 203-206) may improve the compilation success rates and hence may significantly impact the results.
>
> Our current experiments query the models only once per instance for cost reasons. In addition, our query parameters (e.g., temperature=0) limits variance in the output.
>
> We ran a quick experiment that tried to use up to 3 completions with a temperature of 0.7 on GPT-4V for Image2Latex and Image2Music scenarios and keep the first result that compiles. Very few instances that did not compile on the first completion do compile on a later one.
>
> We will report a mini-experiment in the camera-ready manuscript where we will compare the compilation rate of multiple queries against the case of one try on a small subset of data.
>
> > The source image may have secondary information (background, fonts and font size, and other stylistic information) which the VLMs may find hard to extract which may make it difficult to use predefined metrics in a meaningful way.
>
> We acknowledge this limitation of our metrics and stated them in the limitation section of the paper [Lines 267–271]:
> “The automated evaluation in Image2Struct hinges on having good metrics to compare the output image to the input image. The metrics introduced in this paper are not perfect even though they have high correlation with the edit distance between the generated and ground-truth source code. Our work opens up future research directions for the development of methods that can evaluate the degree of similarity between rendered and ground-truth images.”
>
> > Examples of “multiple correct answers” (lines 7-9) should be shared.
>
> We have included examples in the camera-ready version in a new section: Appendix F. Example of multiple correct answers. We replicate them in Section 2 of the attached PDF.
>
> > It will be good to evaluate the metrics themselves on a separately validated, perhaps manually curated, dataset containing ‘correct’ extractions.
>
> Thank you for your suggestion. We will randomly select data points, rate the VLM output manually, and report the results in the camera-ready version.
>
> > Metric distributions for both good code extraction and erroneous ones should be created and compared. This will allow a better utilization (and appropriate improvement) of the metrics for scoring and ranking models.
>
> Thank you for the suggestion. We will include such an analysis in the camera-ready version.
>
> > Was a quality check done after the preprocessing steps?
>
> Yes, we have constantly monitored the quality of the data and tweaked our data collection procedure. In fact, the sampling criterion where only “a maximum of 40 data points per task per day” [Lines 197-198] were collected was added after we detected there was a lack of data diversity. These quality checks will be enhanced and developed further in future works. We will add language to emphasize that we have manually sampled and checked the quality of the sampled test instances in the final draft.
>
> > The definition of Block-EMD is not clear. Is it on a block of the screenshot or something else (‘patches of pixels’ on line 181)?
>
> Thank you for your feedback. Block-EMD first splits up the input image into patches before operating on each of the patches. We have rewritten the section on block-EMD and EMS (see Section 1 of attached PDF). We hope that the descriptions will clear up any misunderstanding.
>
> > The normalization in (2) needs to be clarified.>> Is the final EMS score summed over all the blocks?
>
> Thank you for your feedback. Block-EMD computes a single score for an image and the EMS score is simply a normalized version of it. We hope that the rewritten section addresses this.
>
> > What is the size of the dataset?
>
> We collected a total of 900 instances for webpages (300 each for HTML, CSS and JS), 1200 instances for LaTeX (300 each for equations, tables, algorithms, and plots), and 300 for music for a grand total of 2400 test instances. These numbers are part of the datasheet and we have added the datasheet as an appendix. Additionally, we have added these numbers and other statistics (collection dates, parameters used in filtering and more).

---

> > ### Comment · Reviewer_ksMf · 2024-08-27
> > **Final words**
> >
> > Thanks for spending additional time on taking care of (some of) my original concerns. I have increased my score accordingly. Since this brings the rating in line with other reviewers' rating, there seems to be no need to have a further dialog.
> > All the best.

---

> > > ### Author Response · Authors · 2024-08-27
> > >
> > > Thank you for taking the time to give us helpful comments too.

---

### Decision · Program_Chairs · 2024-09-26

**Decision:**

Accept (Poster)

**Comment:**

This paper introduces the Image2Struct benchmark to evaluate VLMs on the task of extracting structured information from images. The benchmark is designed for automatic closed-loop evaluation of new, unseen data from the real-world that may be continuously/ periodically scraped so that it can’t be ‘gamed’ by VLMs. The benchmark is constructed on three tasks – generation of compilable code from screenshots of webpages (to HTML, CSS, JS code), music sheets (to Lilypond code), or latex algorithms, equations, plots or tables (to latex code). Six SOTA closed-API and six open source VLM models are evaluated. The benchmark shows that VLMs don’t perform well on these tasks in general and that there is significant room for improvement before these models are able to solve these tasks.

Pros: 1) Transparency. The dataset, website and document are currently available and transparent. The information disclosed already shows the important information regarding this benchmark that we may care about. From the materials currently released, it can be used and followed by other researchers in the community. 2) Novelty. The idea of generating images with the extracted code and make comparison with the original images, is quite novel, elegant, and smart. It is different to most of the existing benchmarks with human annotations. 3) Potential of the dataset to keep being updated. As the benchmark does not require human annotation, it becomes easier to keeping it updated, which makes it a fresh and solid benchmark for comparing methods that is difficult to game.

Cons:1) Literature Survey: Important related art is missing. 2) The correctness of the proposed evaluation strategy. If the extracted code contains a few small, forgivable grammatical error that may result in compiling error, does the model fail completely on this case? That seems not reasonable.

In summary, all reviewers agreed that this is very solid submission and authors also handled concerns from reviewers during discussion period. I recommend acceptance.